# TOWARDS GENERAL NEURAL SURROGATE PDE SOLVERS WITH SPECIALIZED NEURAL ACCELERATORS

## ABSTRACT

Surrogate neural network-based partial differential equation (PDE) solvers have the potential to solve PDEs in an accelerated manner, but they are largely limited to systems featuring fixed domain sizes, geometric layouts, and boundary conditions. We propose Specialized Neural Accelerator-Powered Domain Decomposition Methods (SNAP-DDM), a DDM-based approach to PDE solving in which subdomain problems containing arbitrary boundary conditions and geometric parameters are accurately solved using an ensemble of specialized neural operators. We tailor SNAP-DDM to 2D electromagnetics and fluidic flow problems and show how innovations in network architecture and loss function engineering can produce specialized surrogate subdomain solvers with near unity accuracy. We utilize these solvers with standard DDM algorithms to accurately solve freeform electromagnetics and fluids problems featuring a wide range of domain sizes.

## 1 INTRODUCTION

Large scale physics simulations are critical computational tools in every modern science and engineering field, and they involve the solving of parametric partial differential equations (PDEs) with different physical parameters, boundary conditions, and sources. Their ability to accurately capture underlying physical processes makes them particularly well suited in modeling and optimization tasks. Conventionally, PDE problems are set up using discretization methods such as the finite element or finite difference formalisms, which frame the PDE systems as large sparse matrices that are solved by matrix inversion. Problems are set up from scratch every time and computational scaling with domain size is fundamentally tied to the scaling of matrix inversion algorithms.

Neural network-based approaches to solving PDE problems have emerged and have garnered great interest due to their tantalizing potential to exceed the capabilities of conventional algorithms. One of the earliest and most prominent concepts is the Physics Informed Neural Network (PINN), which produces an ansatz for a given PDE problem (Raissi et al., 2019; Karniadakis et al., 2021; Cai et al., 2021). PINNs have been shown to be able to solve wave propagation problems with fixed domain size and domain geometry, but their accuracy is sub-optimal (Moseley et al., 2020; Rasht-Behesht et al., 2022) in systems featuring high spatial frequency phenomena (Wang et al., 2022; Farhani et al., 2022). In addition, they require retraining every time the PDE problem is modified, making them unsuitable for solving generalized parametric PDE problems.

Neural Operators, which are the focus of this study, have also been recently proposed as deep network surrogate PDE solvers. Unlike PINNs, Neural Operators learn a family of PDEs by directly learning the mapping of an input, such as PDE coefficients, to corresponding output solutions using simulated training data. PDE solutions are evaluated through model inference, as opposed to model training, which enables exceptionally high speed PDE problem solving. Initial work on Neural Operator models can be traced to PDE-Net (Long et al., 2018; 2019), and additional improvements in network architecture have been proposed with DeepONet (Lu et al., 2019) and Fourier Neural Operators (FNO) (Li et al., 2020). While much progress has been made, Neural Operators cannot yet directly scale to large arbitrary domain sizes, and they cannot accurately handle arbitrary boundary conditions. These challenges arise due to multiple reasons: 1) the dimensionality of PDE problems grows exponentially with problem scale and can outpace the expressiveness of deep neural networks; 2) it remains difficult to scale neural networks to large numbers of parameters; and 3) the large scale generation of training data for the training of large scale models is resource consuming.

In this work, we propose Specialized Neural Accelerator-Powered Domain Decomposition Methods (SNAP-DDM), which is a qualitatively new way to implement Neural Operators for solving large scale PDE problems with arbitrary domain sizes and boundary conditions. Our method circumvents the issues posed above by subdividing global boundary value problems into smaller boundary value subdomain problems that can be tractably solved with Neural Operators, and then to stitch together subdomain solutions in an iterative, self-consistent method using Domain Decomposition Methods (DDMs). DDMs are the basis for solving large PDE problems with parallel computing (Smith, 1997; Dolean et al., 2015), and they can be implemented using various algorithms including the Schwarz, finite-element tearing and interconnecting (Wolfe et al., 2000), optimized Schwarz (Gander et al., 2002), two-level (Farhat et al., 2000), and sweeping preconditioner (Poulson et al., 2013) methods. While DDM methods have been previously explored in the context of PINNs (Jagtap et al., 2020; Jagtap & Karniadakis, 2021), the accurate solving of arbitrary PDE problems using the combination of Neural Operators and DDM has not been previously reported.

A principal challenge in adapting Neural Operators to DDM is that the subdomain solvers require exceptional accuracy and generalizability to enable accurate DDM convergence (Corigliano et al., 2015). To address this challenge, we train specialized Neural Operators that each solve particular classes of subdomain problems, such as those containing only sources or structural geometric parameters as model inputs. We also propose the Self-Modulating Fourier Neural Operator (SM-FNO) architecture, an augmented FNO architecture with modulation connections that is capable of learning complex PDE boundary value operators with over 99% accuracy. We integrate these Neural Operators directly into a Schwarz DDM iterative framework, where field solutions within each subdomain are iteratively solved until the field solutions in and between every subdomain are self-consistent, at which point the global field solutions are converged.

## 2 METHODS

For this study, we will initially focus on classical electromagnetics (EM) as a model system for detailed analysis, followed by demonstrations of SNAP-DDM to fluid mechanics problems. Classical EM PDEs are governed by Maxwell's equations, from which consolidation of Faraday's and Ampere's laws produce PDEs in the form of wave equations. The magnetic field wave equation is:

$$\nabla \times (\frac{1}{\varepsilon(\mathbf{r})}\nabla \times \mathbf{H}(\mathbf{r})) - \mu_0\omega^2\mathbf{H}(\mathbf{r}) = i\omega\mathbf{J}(\mathbf{r}) \tag{1}$$

$\omega$ is angular frequency, $\varepsilon(\mathbf{r})$ is a heterogeneous dielectric material distribution that is a function of spatial position $\mathbf{r}$, $\mathbf{J}(\mathbf{r})$ is the current source distribution, and $\mathbf{H}(\mathbf{r})$ is the magnetic field distribution to be solved. In two dimensions, the solutions to Maxwell's equations can be reduced to those involving a TE and TM polarization basis. For the TM polarization basis ($H_z, E_x, E_y$), which will be the focus here, equation 1 reduces to a 2D scalar wave equation for $H_z$.

To adapt our surrogate PDE solver to more generalized global simulation frameworks, we consider two types of global boundary conditions. One is the Bloch periodic boundary condition, in which the fields at one boundary wrap around and connect with field solutions at the opposite boundary with an additional Bloch phase. These boundaries are used to model infinitely periodic systems. The second is the perfectly matched layer (PML), which have complex valued, spatially varying dielectric constants that are specified to absorb incident electromagnetic waves without backreflection. Various forms of PMLs have been proposed, and we utilize the uniaxial PML (Gedney, 1996). The Bloch periodic and PML boundaries can be used exclusively for both horizontal and vertical-orientated global boundaries or be used in combination to model semi-infinite scattering systems.

The global simulation domain is parameterized using the finite difference formalism, in which the electromagnetic fields, sources, dielectric material distributions, and boundary conditions are specified on a rectilinear grid using the Yee formalism (Yee, 1966). In this manner, the discretized field solutions are stable and consistent with the continuous differential form of Maxwell's equations and Equation 1. In the frequency domain, training data is generated using established finite-difference frequency domain (FDFD) solvers (Hughes et al., 2019). For this study, we consider global electromagnetic simulation domains that are discretized with pixel dimensions of $6.25 \times 6.25$ nm. The source wavelength is 1050 nm and the dielectric materials have a refractive index ranging from 1 to 4. Our consideration of dielectric media featuring relatively high dielectric constant values presents unique challenges in modeling due to their ability to support strongly resonant optical interactions.

## 2.1 SNAP-DDM

We consider a domain decomposition approach in which we subdivide the global domain into overlapping square subdomains with fixed $64 \times 64$ pixel dimensions (Figure 1a). Each subdomain for a given DDM iteration poses a PDE boundary value problem with magnetic field boundary conditions that are solved using a specialized pretrained subdomain model. In this study, we utilize the overlapping Schwarz DDM formalism using Robin boundary conditions for each subdomain solver (Figure 1b), also known as the transmission Robin type boundary condition, which is used in optimized Schwarz Methods (Gander et al., 2001; Dolean et al., 2009). The DDM iterative algorithm is illustrated in Figure 1c and the procedure is summarized as follows:

1. The magnetic field is initialized to be zero everywhere.

2. For the $k^{th}$ iteration, a boundary value problem is solved at each subdomain using a specialized subdomain model. The inputs to each subdomain model are the subdomain Robin boundary values and a specialized image of the domain (i.e., image of the material, source, or global boundary) and the output is an H-field map.

3. H-fields outputted from the subdomain models are used to update the subdomain Robin boundaries in all subdomains, which are used as model inputs for the $(k + 1)^{th}$ iteration.

4. The algorithm terminates when a predetermined number of iterations is executed or when the physics residue falls below a predetermined threshold.

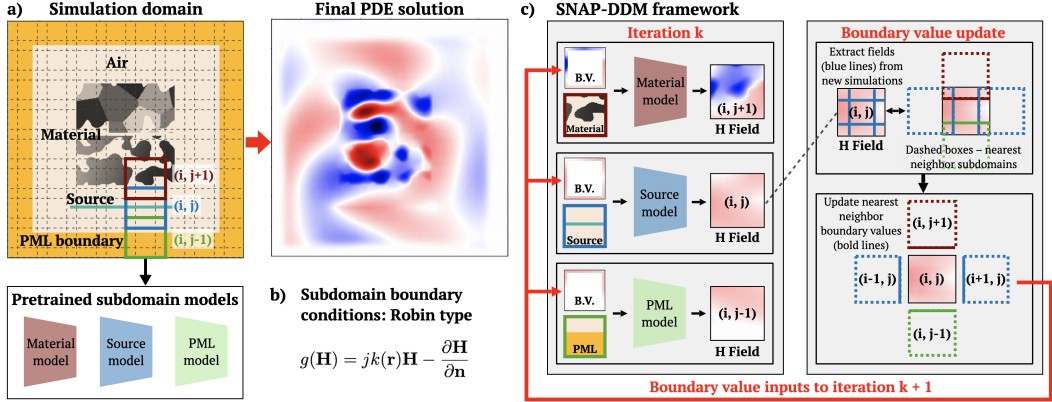

Figure 1: SNAP-DDM framework. a) Global simulation domain and corresponding H-field solution for a 2D electromagnetics problem featuring arbitrary sources, global boundary conditions, and freeform grayscale dielectric structures. The global domain is subdivided into overlapping subdomains parameterized by position $(i, j)$. Three types of specialized Neural Operator models are trained to solve for three types of subdomain problems. b) Expression for the Robin type boundary condition used in the specialized Neural Operator subdomain models. $k(\mathbf{r}) = 2\pi\varepsilon(\mathbf{r})/\lambda$ is the wave vector in a medium with dielectric constant $\varepsilon$ and $\mathbf{n}$ is the outward normal direction. c) Flow chart of the iterative overlapping Schwarz method. In iteration $k$, electromagnetic fields in each subdomain are solved using the specialized Neural Operators, and the resulting fields are used to update the subdomain boundary value inputs for iteration $k + 1$. The "Boundary value update" box shows how solved fields in the $(i, j)$ subdomain are used to update the boundary value fields in nearest neighbor subdomains for subsequent DDM iterations. **B.V.**: boundary value. **PML**: perfectly matched layers.

It is essential that the trained subdomain PDE surrogate solvers have near unity accuracy to ensure that the DDM algorithm accurately converges. We introduce two innovations to enhance solver accuracy. First, we train specialized neural operators that each solve specific classes of PDE problems. For 2D EM problems, we consider three types of Neural Operators that each specialize in solving: 1) subdomains containing only PMLs in air; 2) subdomains containing only sources in air; and 3) subdomains containing only heterogeneous grayscale material and air structures. Compared to a generalized surrogate subdomain solver that would require many data inputs, our specialized subdomain solvers each have inputs consisting of only a single specialized image together with a vector containing the magnetic field Robin boundary conditions, which reduces the dimensionality

of the learning problem and enables specialized physics to be more accurately captured in each network. Additional specialized neural operators can be considered with increasing problem domain complexity without loss of generality.

Second, we modify the original FNO architecture (Li et al., 2020) and introduce the Self-Modulating Fourier Neural Operator (SM-FNO) subdomain surrogate solver (Figure 2). We specifically incorporate two key features, the first of which we term a modulation encoder. Mechanistically, we utilize multiple residual blocks and fully connected layers to compress the input data into a latent modulation tensor, which then modulates each linear transformation $\mathbf{R}$ in the neural operator through element-wise multiplication (Figure 2). This concept builds on our observation that in the original FNO architecture, the linear transform weight $\mathbf{R}$ in each Fourier layer is fixed and are independent of the network input parameters, limiting the ability of the neural operator to accurately process the highly heterogeneous input data featured in our problem. This modification is inspired by the efficacy of self-attention in transformer architectures (Vaswani et al., 2017) and multiplicative interactions between inputs in PINNs (Wang et al., 2021). We show in Section 4 that the modulation method we introduced is crucial to enhance the expressivity of the model.

The second feature we propose is the explicit addition of a residual connection within each Fourier layer. The residual connection concept dates back to the ResNet architecture (He et al., 2016), where such connections were shown to mitigate the vanishing gradient problem and enable the training of deeper models. From our experiments, we have discovered that explicit residual connections are necessary for training deeper FNOs, especially when inputs are augmented with auxiliary data like boundary values. We note that the residual connection is equivalent to initializing the $1 \times 1$ convolutional layer $\mathbf{W}$ using identity plus Kaiming or Xavier initialization (He et al., 2015), but we keep the residual connection in Figure 2 for clarity and ease of implementation. We also note that related concepts involving the addition of residual connections to the FNO architecture have been explored elsewhere (Tran et al., 2021).

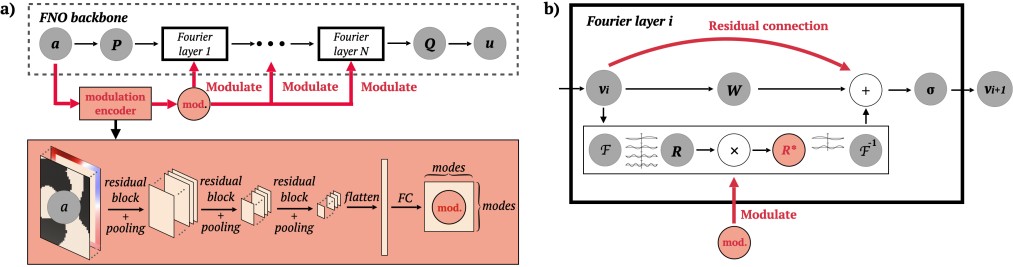

Figure 2: Self-Modulating Fourier Neural Operator Architecture for DDM subdomain solvers. Modifications to the standard Fourier Neural Operator (FNO) are highlighted in red and include: 1) the addition of self-modulation connections that encode the input into a tensor, which is then multiplied with the linear transformation matrix $\mathbf{R^*}$ in each Fourier layer; and 2) the addition of residual connections inside each Fourier layer. $\mathbf{a}$: network input comprising a stack of images specifying the specialized subdomain layout and Robin boundary values. $\mathbf{u}$: network output comprising a stack of images specifying the real and imaginary H-field maps. $\mathbf{P}$: fully connected layer that increases the number of channels. $\mathbf{Q}$: fully connected layer that decreases the number of channels. $\mathcal{F}$: Fourier transform. $\sigma$: leaky-ReLU activation function (Xu et al., 2015). $\mathbf{W}$: channel mixer via 1-by-1 kernel convolution. $\mathbf{R}$: original linear transform on the lower Fourier modes. $\mathbf{R^*}$: modulated linear transform through an element-wise multiplication with the modulation tensor.

## 3 EXPERIMENTS

### 3.1 NETWORK TRAINING AND BENCHMARKING

To curate the training data, we build a data generation platform based on 1000 full-wave simulations, each containing heterogeneous dielectric distributions and sources within a $960 \times 960$ pixel simulation domain. The structural layouts are specified by a pipeline inspired from image processing, with details provided in Appendix A. Random magnetic current sources surrounding the devices are

specified as a superposition of sinusoidal functions with random amplitudes and phases. The resulting simulated fields are then cropped into $64 \times 64$ pixel sections to produce the subdomain training dataset. Using this approach, we generate a total of 1.2M subdomain training data samples (100k for the PML solver, 100k for the source solver, and 1M for the grayscale material solver). To train the subdomain solvers, we apply a training scheme that utilizes a hybrid data-physics loss function:

$$L = L_{data} + \alpha(L_{pde} + cL_{bc}) \tag{2}$$

Detailed expressions of the data and physics loss terms are in Appendix B. $c$ is a constant weight set to 1 for simplicity (we found the model performance is insensitive to its value between $0.1$ and $10$) and $\alpha$ is a dynamic weighting term that balances data loss and physics loss. Regularization of network training with physical loss serves to explicitly enforce physical relationships between nearest neighbor field pixels, which enhances the accuracy of magnetic field spatial derivatives calculated using the finite differences method (Chen et al., 2022b). Such accuracy is critical for evaluating electric fields and Robin boundary conditions from inferenced magnetic fields.

We benchmark our trained SM-FNO subdomain solver with the UNet (Ronneberger et al., 2015), Swin Transformer (Liu et al., 2021) (details in Appendix F), the classical FNO, and the recently improved version of FNO termed F-FNO (Tran et al., 2021). We also train our SM-FNO solver without physics loss. The networks are trained with both 100K and 1M total data to show their dependency on data scaling, except for the Swin Transformer model, which is only benchmarked on 100k training data (training on 1M data would take 2 months). The 100k and 1M data are split into 90% training data and 10% test data. All models use a batchsize of 64 and are trained for 100 epochs for 100k training data or 50 epochs for 1M training data. The Adam optimizer with an individually fine-tuned learning rate is used with an exponential decay learning rate scheduler that decreases the learning rate by $30\times$ by the end of training. A padding of 20 pixels is applied to all FNOs.

We consider only the material subdomain model in this analysis for simplicity and specify architectures with similar floating point operations (FLOP) and model weights. All data in Table 1 and plots in Figure 3 are conducted on 10k newly generated, unseen test data. We see that the specification of a targeted model architecture is crucial to achieving high accuracy. The vanilla FNO fails to learn the problem with good accuracy, even with a large number of model weights. While the Swin transformer requires relatively fewer neural network weights, the expensive self-attention operations require over 10x FLOPs compared to FNO-based architectures. A comparison of SM-FNO-data-only and SM-FNO indicates that the explicit inclusion of Maxwell's equations leads to a dramatic reduction of $L_{pde}$ and $L_{bc}$, which is essential to getting the DDM algorithm to converge. Our largest model, SM-FNO-v2, is 99.0% accurate and features exceptionally low $L_{pde}$ and $L_{bc}$.

Table 1: Electromagnetics: Subdomain model benchmark on 10k test data

| Model (trained on 100k data) | $L_{data}$ (%) | $L_{pde}$ (a.u.) | $L_{bc}$ (a.u.) | Param (M) | FLOP (G) |
|---|---|---|---|---|---|
| FNO-v1 | 9.04 | 2.21 | 0.309 | 73.8 | 0.79 |
| F-FNO-v1 | 8.32 | 1.69 | 0.163 | 4.9 | 1.45 |
| UNet-v1 | 5.40 | 0.73 | 0.099 | 5.2 | 1.53 |
| Swin T-v1 | 5.15 | 2.15 | 0.148 | **1.9** | 9.60 |
| SM-FNO-v1-data-only(ours) | 3.95 | 7.08 | 0.162 | 4.7 | **0.66** |
| SM-FNO-v1(ours) | **3.85** | **0.50** | **0.067** | 4.7 | **0.66** |
| Model (trained on 1M data) | $L_{data}$ (%) | $L_{pde}$ (a.u.) | $L_{bc}$ (a.u.) | Param (M) | FLOP (G) |
| FNO-v2 | 5.34 | 1.43 | 0.124 | 131.2 | 1.86 |
| F-FNO-v2 | 3.52 | 0.84 | 0.078 | 13.3 | 2.59 |
| UNet-v2 | 2.93 | 0.44 | 0.080 | 11.1 | 3.28 |
| SM-FNO-v2-data-only(ours) | 1.36 | 2.76 | 0.073 | **10.2** | **1.43** |
| SM-FNO-v2(ours) | **1.01** | **0.30** | **0.030** | **10.2** | **1.43** |

Model FLOPs are computed using the open-source library *fvcore*. The FLOPs of FFT operations are computed using the formula: $2L(NC^2 + NC\log N)$ for 2d FFTs, and $2L((H + W)C^2 + NC(\log H + \log W))$ for 1d FFTs, in which $N = HW$ is the number of pixels of each channel,

$L$ is number of layers and $C$ is number of channels (Guibas et al., 2021). The factor 2 accounts for forward and inverse FFT operations.

Figure 3: Benchmarking of material boundary value subdomain solvers on unseen test data. The model inputs are a grayscale material dielectric distribution image ($\varepsilon = 1$ to $\varepsilon = 16$) and Robin boundary conditions, and the outputs are images of the real and imaginary H-fields. The real parts of outputted H-fields are shown. The L1 data loss is normalized to the mean absolute ground truth field value and the physics residue map is the summed expression in Equation 5.

## 3.2 LARGE SCALE ELECTROMAGNETICS SIMULATIONS

Large scale electromagnetic simulations comprising high contrast heterogeneous media are notoriously hard to solve using end-to-end neural surrogate solvers. We show in the Supplementary Section that the training of a Fourier neural operator to solve full-scale problems leads to fundamental scaling bottlenecks in dataset size, model size, and memory usage. We also find that PINNs struggle to scale up to large simulation domains comprising high dielectric contrast media, and that the solutions produced from trained PINN models are particularly sensitive to their detailed initialization and training conditions. These results are consistent with recent large scale simulation demonstrations in the literature: one concept based on graph networks featured errors of 28% (Khoram et al., 2022) and another concept based on neural operators featured errors ranging from 12% to 38% (Gu et al., 2022).

On the other hand, our SNAP-DDM algorithm combining trained subdomain surrogate solvers with the overlapping Schwartz DDM method produces a qualitatively different and better result. To demonstrate, we solve a variety of large scale electromagnetics problems featuring a wide range of dielectric constant and domain size configurations. We use the SM-FNO-v2 architecture for the material and PML models and the lighter SM-FNO-v1 network for the source model. For each problem, the global domain is initially subdivided into an array of subdomains, each of which are classified as PML, source, or material subdomains. During each SNAP-DDM iteration, data from subdomains of a given class are aggregated into a batch and inputted into the corresponding specialized SM-FNO, which infers and outputs the H-field solutions of the batch in a parallelized manner. The DDM algorithm is stopped after a predetermined number of iterations.

Representative electromagnetic simulation results are shown in Figure 4 and demonstrate the versatility and accuracy of SNAP-DDM. The simulations feature widely varying global domain sizes, indicating the ability for our scheme to readily adapt to arbitrary global domain sizes through the tiling of different numbers of subdomains and tailoring the amount of overlap between subdomains. Some of these simulations feature the use of PML boundaries on all sides, which is ideal for purely

scattering simulations, while others comprise half PML and half Bloch periodic boundaries, which are a natural boundary choice for semi-periodic systems. The off-normal incident field in the thin film problem is achieved by tailoring the line source profile with the appropriate Bloch phase. For all of these examples, the final and ground truth fields appear indistinguishable, and the absolute error in the final fields in all cases is near 5%.

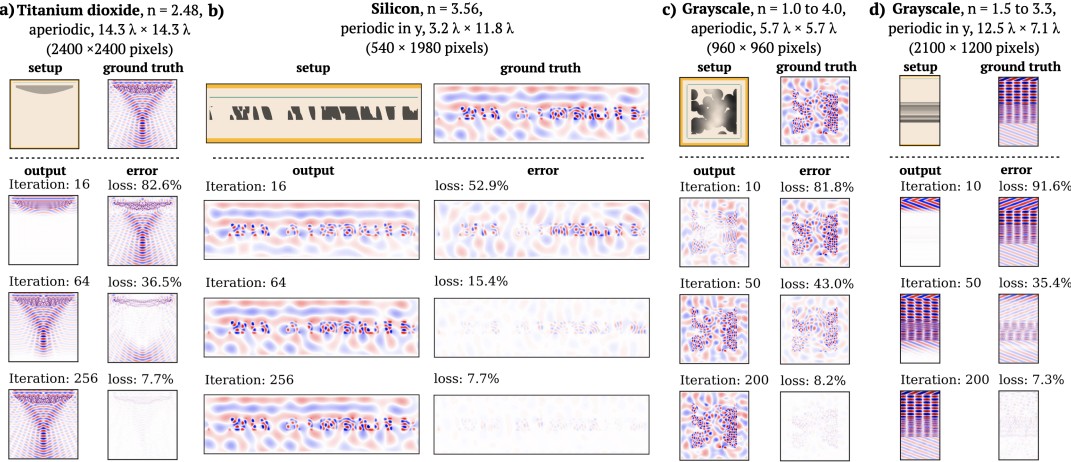

Figure 4: SNAP-DDM evaluated on different electromagnetics systems. These systems include: **a)** a titanium dioxide microlens, **b)** a thin film silicon-based metasurface, **c)** a volumetric grayscale metamaterial scatterer, and **d)** an optimized grayscale thin film stack featuring high reflectivity. The simulation domain contains pixels with physical dimensions of $6.25nm$ and the wavelength $\lambda = 1.05\mu m$. Ground truth fields, model output fields, and field error are plotted on the same scale for each device. The largest simulation domain is in (a) and comprises an array of $40 \times 40$ subdomains.

### 3.3 TIME COMPLEXITY

In this section, we benchmark the time complexity between SNAP-DDM and a conventional FDFD solver for 2D EM problems. Based on the current accuracy level of SNAP-DDM, we choose to benchmark the time it takes to reach an average mean absolute accuracy of 15%, evaluated on 10 random devices for each domain size. For SNAP-DDM, the number of iterations required for convergence strongly depends on the material refractive index, and we therefore benchmark computation time for simulation domains containing either silicon dioxide (n = 1.5) or titanium dioxide (n = 2.48). Square domains with sizes ranging from $600 \times 600$ pixels to $2100 \times 2100$ pixels are evaluated. The FDFD benchmark simulations are performed on 10 grayscale dielectric structures for each domain size with refractive indices ranging from those of silicon dioxide to titanium dioxide. SNAP-DDM is run with one NVIDIA RTX A6000 GPU, and the FDFD solver runs on single CPU of model Intel Xeon Gold 6242R. From the time benchmark (See Appendix D), we observe that for silicon dioxide, which has a relatively low refractive index, SNAP-DDM has better performance compared to the FDFD solver. However, for titanium dioxide, SNAP-DDM requires significantly more iterations and ultimately takes a much longer time than FDFD.

### 3.4 SNAP-DDM STEADY STATE FLUID FLOW SIMULATIONS

The SNAP-DDM concept can apply to a broad range of steady state PDE problems, and we demonstrate here the application of SNAP-DDM to 2D steady state fluid flow problems. Fluid mechanics systems are governed by the incompressible Navier-Stokes (NS) equation, which is:

$$\frac{\partial \boldsymbol{u}}{\partial t} + (\boldsymbol{u} \cdot \nabla)\boldsymbol{u} - \nu\nabla^2\boldsymbol{u} = -\frac{1}{\rho}\nabla p \qquad (3)$$

We consider steady state flows in an arbitrary-shaped pipe with circularly shaped obstacles and a viscosity of $\nu = 0.08$ (Chen & Doolen, 1998). To train our subdomain boundary value solvers

for these problems, we first simulate flows using the time domain Lattice-Boltzmann Method and run the simulations until the flows are at steady state. A total of 200 ground truth simulations with $900 \times 600$ pixel domain sizes are generated, from which the data for 100k subdomains with $64 \times 64$ pixel sizes is produced to form the subdomain training dataset. Further details are in Appendix A. The subdomain model takes an image of the obstacle and velocity field $(u, v)$ Robin boundary conditions as inputs, and it outputs images of the full velocity field. To compute physics loss, ground truth pressures are used with the steady state version of Equation (3).

Benchmark results of our SM-FNO subdomain solver are summarized in Table 2 and Figure 5a, and they indicate that our SM-FNO network displays the lowest data and physics loss compared to alternative subdomain solver architectures, with data loss approaching 1%. Demonstrations of steady state fluid flow simulations with SNAP-DDM are shown in Figure 5b, where the simulations produce velocity profiles with errors less than 15%. The reduced accuracy in the flow SNAP-DDM simulations, compared to those from electromagnetics, is likely due to the sub-optimal performance of Schwartz DDM with Robin boundary conditions for steady state flow problems. DDM for fluids problems continues to be a topic of active research, and further improvement in SNAP-DDM for fluids will be followed up in future work.

Table 2: Steady state flow: subdomain model benchmark on 10k test data

| Model | $L_{data}$ (%) | $L_{pde}$ (a.u.) | $L_{bc}$ (a.u.) | Param (M) | FLOP (G) |
|---|---|---|---|---|---|
| FNO | 6.5 | 1.77 | 2.44 | 69.2 | 0.34 |
| Swin T | 5.1 | 1.10 | 0.13 | **1.9** | 9.60 |
| UNet | 1.8 | 0.35 | 0.10 | 5.2 | 1.53 |
| SM-FNO(ours) | **1.3** | **0.23** | **0.06** | 4.7 | **0.66** |

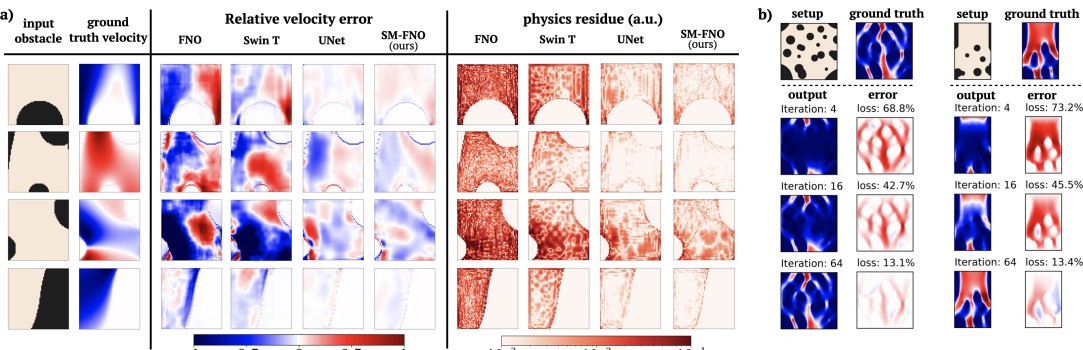

Figure 5: SNAP-DDM pipeline for steady-state fluid flow problems. **a)** Subdomain model benchmark. **b)** Steady state fluid flow velocity fields solved with SNAP-DDM. Subdomain grid sizes are $8 \times 8$ (left) and $8 \times 5$ (right).

## 4 ABLATION STUDY

We perform an ablation study to understand the contribution of each modification to the FNO featured in our SM-FNO. Results are shown in Table 3 where L is the number of layers, C is the number of channels (hidden dimension), and M is the number of Fourier modes for linear transform. We start with the vanilla FNO without residual connections and find that the model fails to consistently learn for depths larger than 4 layers. Upon training with 4 layers, increases in hidden dimension and number of modes increases model size without contributing much to performance. When self modulation is removed from the SM-FNO, deeper networks could be trained with the residual connection but different depths and widths produced similar sub-optimal performance. This indicates that without the modulation path, model expressivity is limited. When we remove the residual connections from the SM-FNO, the model did not work well with large depths. The best model contained 4 layers and produced reasonable accuracy.

It is clear the two modifications that we added to the FNO architecture are both synergistic and essential to improving subdomain solver accuracy: the residual connection enables deep architectures to be trained while the self-modulation connection increases the model expressivity by promoting self-multiplicative interactions within each input. In addition, the hybrid physics-augmented training scheme significantly lowers the physical residue while slightly reducing the data loss. We also point out that our use of Robin boundary conditions ensures that the subdomain solvers solve a well-posed PDE problem, unlike alternative boundary conditions such as Dirichlet boundary conditions.

Table 3: Ablation study

| Model | $L_{data}$ (%) | $L_{pde}$ (a.u.) | $L_{bc}$ (a.u.) | Param (M) | FLOP (G) | L | C | M |
|---|---|---|---|---|---|---|---|---|
| FNO without residual connection | 16.01 | 1.92 | 0.217 | 41.0 | 0.97 | 4 | 100 | 16 |
| SM-FNO remove residual connection | 7.61 | 1.03 | 0.123 | 8.9 | 0.78 | 4 | 44 | 16 |
| SM-FNO remove self modulation - 1 | 8.04 | 1.93 | 0.166 | 42.0 | 1.00 | 10 | 64 | 16 |
| SM-FNO remove self modulation - 2 | 6.11 | 3.29 | 0.155 | 32.8 | 0.82 | 20 | 40 | 16 |
| SM-FNO-v1 | 3.85 | 0.50 | 0.067 | 4.7 | 0.66 | 16 | 16 | 16 |
| SM-FNO-v2-data-only | 1.36 | 2.76 | 0.073 | 10.2 | 1.43 | 16 | 24 | 16 |
| SM-FNO-v2 | 1.00 | 0.30 | 0.030 | 10.2 | 1.43 | 16 | 24 | 16 |

The accuracy of the SNAP-DDM framework is dependent on multiple factors. Plots of DDM accuracy versus iterations for 20 devices under different setups is shown in Figure 6. These plots show that error from SNAP-DDM increases significantly when we replace the large material model with a lighter version, indicating the need for the specialized subdomain models to have near unity accuracy. These plots also show that when a fraction of models is trained without physics, SNAP-DDM error also increases, indicating the need for hybrid data-physics training for all subdomain models.

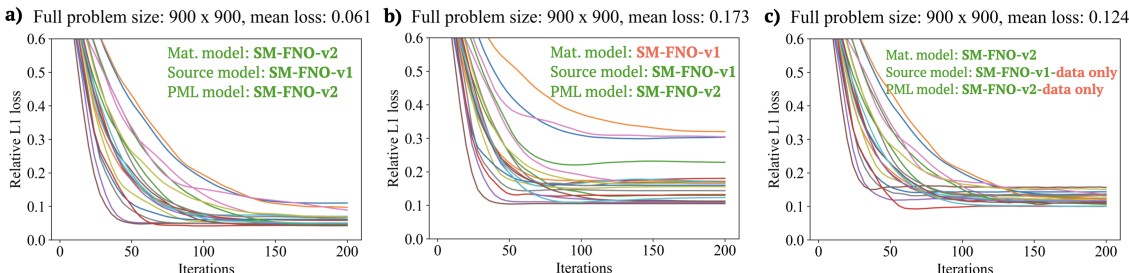

Figure 6: SNAP-DDM convergence curves with different types of subdomain solvers. The plots show DDM algorithm error versus iteration count, with each curve representing field error from a simulated random grayscale device within a $900 \times 900$ pixel domain (15 by 15 subdomains). **a)**: The proposed setup with 3 specialized subdomain models. **b)** Same as (a) but use of a lighter material model trained on 100k data. **c)** Same as (a) but use of source and PML models trained using only data loss. The slower convergence curves correspond to materials with higher average dielectric constant.

## 5 LIMITATIONS AND FUTURE WORK

There are multiple potential speedup strategies with SNAP-DDM that can be considered in future work. Preconditioning is a common strategy in DDM for improving convergence speed by reducing the condition number of the system, especially for large and ill-conditioned problems (Vion & Geuzaine, 2014; Gander & Zhang, 2019). Increasing the subdomain size, and more generally incorporating a non-uniform grid or mesh-to-grid approach to subdomain solving, has the potential to introduce computational savings, though a further quantitative analysis into the balance between model size and latency is required. More sophisticated SNAP-DDM implementations may incorporate multi-level or multi-grid concepts for initialization and improved memory management. We also anticipate that higher order boundary conditions can help with DDM convergence and accuracy.

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

## A  DATA GENERATION

To generate random geometric distributions for 2D EM problems that include both regular-shaped objects and structures with free-form topologies, we adopt a pipeline inspired from image processing:

1. Generate random noise map in range $(0, 1)$.
2. Threshold by some level between $0$ and $1$.
3. Erode the map with a Gaussian filter.
4. Dilate the map using filters with tilted elliptic profiles.
5. Apply Gaussian filter for smoothing.

By tuning the threshold level, filter weight, and erosion and dilation parameters, random shaped geometries with different feature size distributions are generated. We generate Gaussian random fields and Voronoi diagrams with grayscale values between 1 and 16, and then we use the random geometries as masks to create the grayscale material distributions used to produce ground truth data. The Gaussian random field creates a continuously changing dielectric that are representative of features appearing in freeform metamaterial designs. The Voronoi diagram creates boundaries between different constant regions that produces material boundary features in the training data.

Line sources with a mix of random sinusoidal profiles are placed on all four sides of the generated grayscale material to create randomly scattering fields in all directions. PML boundaries are places on all four sides with thickness of 40 pixels. We use ceviche FDFD solver to generate 1000 fullwave simulations of size 960 by 960 pixels, from which we cropped the fields and physical properties to produce an 100k material dataset, an 1M material dataset, an 100k source dataset and an 100k PML dataset for training specialized subdoamin models.

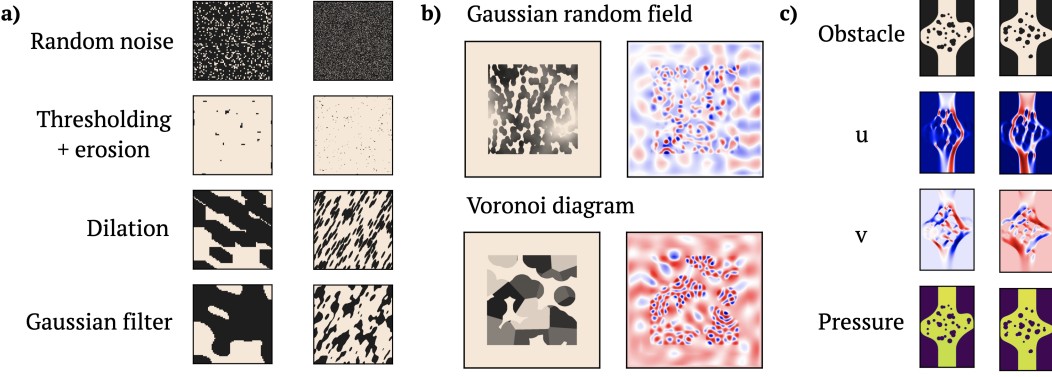

Figure 7: Pipeline for data generation. **a)** Image processing-inspired procedure for generating random dielectric geometries. **b)** Two kinds of grayscale geometries and corresponding simulated H-fields. **c)** Flows in arbitrarily-shaped pipe with circular obstacles.

For steady state fluids simulations, we generate pipes with arbitrarily-shaped middle sections that are created by connecting two random Bézier curves. The idea is to create boundaries beyond straight vertical walls that may appear in freeform flow flow scenarios. Circular obstacles with randomly sampled radii are placed in the pipe in a manner where a minimum gap size is guaranteed. We use constant velocity($u = u_0$, $v = 0$) as the boundary condition for the inlet surface. For the bottom outlet, $v = 0$ and $P = constant$ is used as the boundary condition. The viscosity is fixed to be 0.08 and steady state flow is reached when the relative velocity change after 100 time steps is less than $10^{-4}$. The steady state solution is not guaranteed in this way, but we found that 199 out of 200 simulations reached steady state. For both EM and fluids cases, subdomain data is produced by cropping data from large-scale simulations with optional rotation as data augmentation method.

## B LOSS FUNCTIONS

Here we present the data-physics loss function used to train the subdomain models:

$$L_{data} = \frac{1}{N} \sum_{n=1}^{N} \left\| \mathbf{H}^{(n)} - \hat{\mathbf{H}}^{(n)} \right\|_1 \tag{4}$$

$$L_{pde} = \frac{1}{N} \sum_{n=1}^{N} \left\| \nabla \times \left( \frac{1}{\varepsilon(\mathbf{r})} \nabla \times \mathbf{H}^{(n)} \right) - \mu_0 \omega^2 \mathbf{H}^{(n)} \right\|_1 \tag{5}$$

$$L_{bc} = \frac{1}{N} \sum_{n=1}^{N} \left\| g - \left( jk(\mathbf{r}) \mathbf{H}^{(n)} - \frac{\partial \mathbf{H}^{(n)}}{\partial n} \right) \right\|_1 \tag{6}$$

in which $\hat{\mathbf{H}}$ is the ground-truth magnetic field, $k(\mathbf{r}) = 2\pi\varepsilon(\mathbf{r})/\lambda$ is the wave vector in the medium.

## C HYBRID DATA-PHYSICS TRAINING SCHEME

The subdomain solvers are trained using a hybrid loss function composed of a data term, $L_{data}$, and a physics loss term, $L_{physics}$, which is scaled by a hyperparameter $\alpha$:

$$L = L_{data} + \alpha \cdot L_{physics}.$$

Previous work in physics-augmented neural network training has demonstrated that training convergence and performance is sensitive to the *relative* magnitude of the physics loss term compared to the data loss term (Chen et al., 2022a). To maximize the generality of the proposed training setup, we employ a dynamically tuned hyperparameter, $\alpha$, which is scaled throughout the training process, like the approach taken in the WaveY-Net study (Chen et al., 2022a). At the end of each epoch, $\alpha$ is modified such that the ratio between $\alpha \cdot L_{physics}$ and $L_{data}$ is a constant, $\alpha'$, throughout the entire training process. The practice of dynamically tuning the physics loss coefficient greatly stabilizes training convergence across different simulation problems, thereby allowing a working training scheme to readily generalize to problems governed by different physics equations.

The constant physics ratio, $\alpha'$, is neural network model dependent and appears insensitive to the two types of problems being simulated. All the FNO models are trained with $\alpha' = 0.3$ and the U-Net and Swin Transformer are trained with $\alpha' = 0.1$. However, due to the slower learning rate of vision transformers compared to convolutional neural networks, $\alpha'$ is set to 0 for the first 50 training epochs to prevent divergent training behavior. Divergent behavior occurs in domains with high contrast material due to the presence of strong optical resonances. Without substantial influence from data to push the optimization in regimes with the correct resonances, unstable interplay between bulk and boundary physics loss leads the optimization process astray.

## D TIME COMPLEXITY

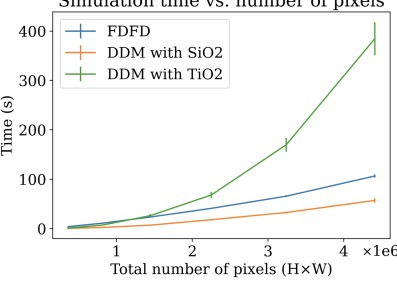

Figure 8: Time complexity comparison with 2D electromagnetics SNAP-DDM and a conventional FDFD solver for two different dielectric materials.

# E    IMPLEMENTATION DETAILS FOR U-NET

The U-Net architecture employed in this study is constructed as follows:

- Each half of the U-shaped architecture contains 5 blocks of convolutional layers;
- Each convolutional block is composed of 6 convolutional layers;
- Each convolutional layer is composed of a sequence of operations: convolution, batch normalization, and ReLU activation;
- The number of convolutional kernels contained in the convolutional layers of each block is increased by a factor of two compared to the previous block's number, starting with 30: $30 \cdot (2^{(block-1)})$. The number in the blocks of the second half of the U-shaped architecture mirror those in the first half.

# F    IMPLEMENTATION DETAILS FOR SWIN TRANSFORMER

We designed our implementation of the Swin Transformer with the same shifting windows and windowed attention as featured in previous architectures (Liu et al., 2021). Notably, we abstain from utilizing patch merging, as our network design maintains consistent input and output dimensions. Our architecture comprises multiple stacked Swin Transformer layers, all configured with uniform patch sizes. The hyperparameters regarding the model is chosen based on the Swin transformer model used in image semantic segmentation tasks, and adjusted based on the input size of our subdomain problem.

To elaborate on the architectural parameters:

- Patch Size: We employ a patch size of 1.
- Window Size: Each attention window spans 9 patches.
- Number of Heads: Multi-head attention is applied with 16 attention heads.
- Swin Transformer Blocks: Each layer of the network contains 16 Swin Transformer blocks with window shifting enabled.
- Layers: The network is formed by stacking 4 such layers.

We initialize trainable absolute positional encodings drawn from a normal distribution with a mean of 0 and a standard deviation of 0.01. For encoding domain-specific information, such as the input refractive index in EM simulations or obstacles in fluid simulations, we employ a matrix multiplication to transform these data into a 48-dimensional vector. Boundary conditions are treated similarly but encoded using a separate encoder. These encodings are then padded around the original image for 4 times, leading to an overall 72 by 72 input to the network. Corners not covered are left as 0. The output of the network is subsequently transformed using another matrix multiplication to ensure it conforms to the dimensions required for the final output. For training the model, we use Adam optimizer with learning rate set to 0.001 for initial, and then exponentially decay it to 0.0001 over the training course of 50 epochs. We do not use dropout or weight decay during training.

