# Supplementary Information:
# Towards general neural surrogate PDE solvers with specialized neural accelerators

## S1 Domain-Specific Physics-Informed Neural Networks Benchmark

In this section, we consider a broader comparison with physics-informed baselines by evaluating the performance of PINNs on the same simulation setup illustrated in Figure 1 of the main text. Physics-informed PDE solvers can be categorized into domain-specific solvers Raissi et al. (2019; 2017a;b) and operator (general function-to-function) solvers Li et al. (2023); Wang et al. (2021). In this section, we benchmark physics-informed domain-specific solvers on full-sized problems. Benchmarks with operator solvers are demonstrated in Section S2.

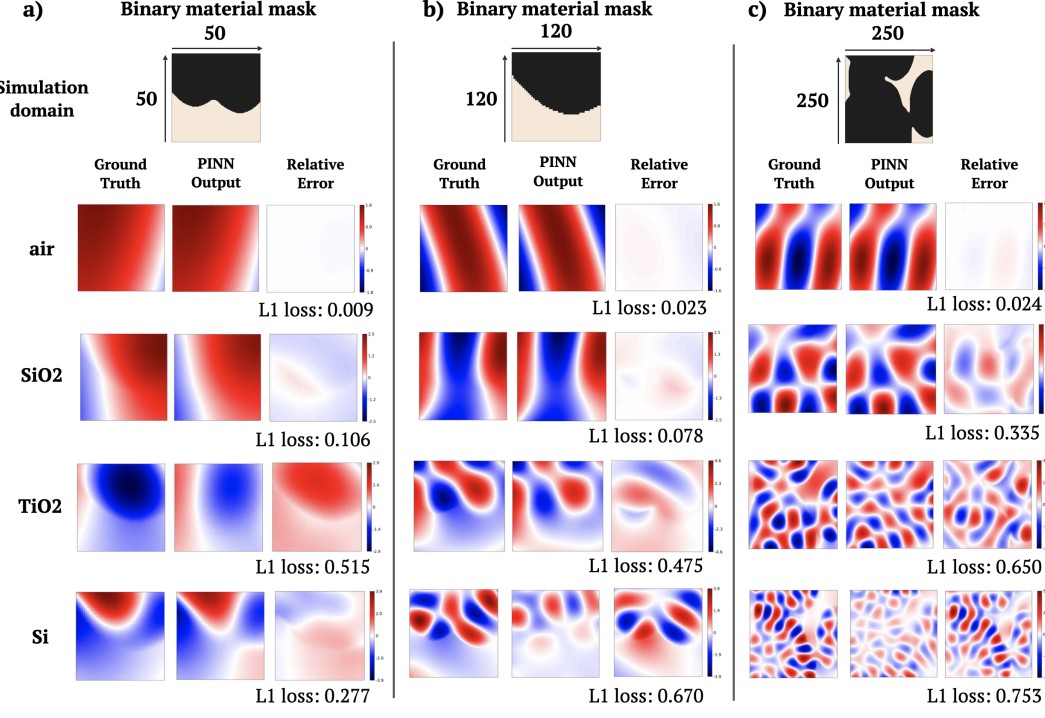

Figure 1: PINNs trained on full-sized problems. Here we show the results of training PINNs with sin activation functions on different problem size and material type. Three problem sizes are investigated: **a)** 50 by 50. **b)** 120 by 120, **c)** 250 by 250. For each problem size, a binary mask defines the material geometry, and four different materials with the same geometry are used to construct 4 different problems. We conduct thorough parameter and architecture search within each problem, and the L1 loss for the best models are reported.

We benchmarked a fully connected PINN architecture with sine activation (Song et al., 2022) on a total of 12 full-sized problems: {air, SiO2, TiO2, Si} × {50 by 50, 120 by 120, 250 by 250}. The simulation domain contains heterogeneous material (material and air),

| Hyperparameter | Values |
|---|---|
| Starting Learning Rate | min: 1e-5 
 max: 1e-3 |
| ADAM Weight Decay | min: 1e-8 
 max: 1e-3 |
| Activation Multiplier (w0) | min: 1 
 max: 30 |
| No. Hidden Layers | min: 2 
 max: 5 |
| Layer Height | {32, 64, 128, 256} |
| B.C. Weight | min: 1 
 max: 20 |

Table 1: Range of possible values during hyperparameter sweep

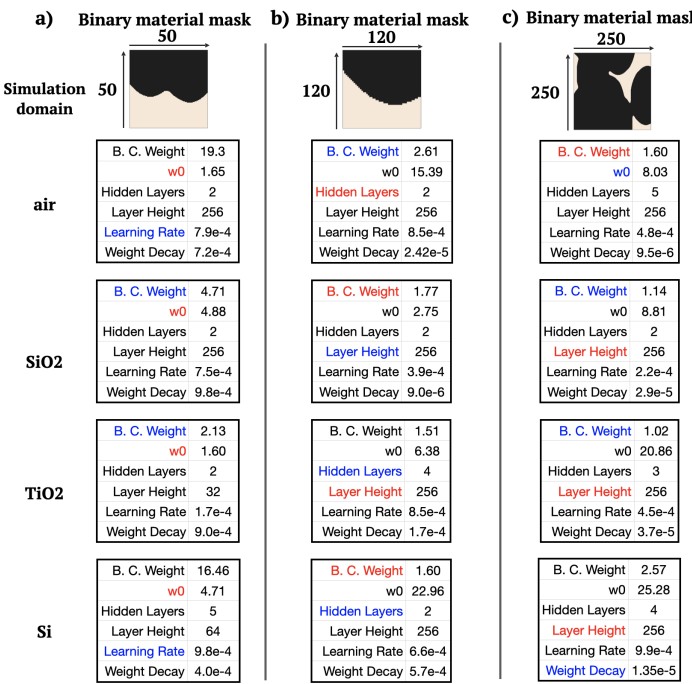

Figure 2: Optimal training hyperparameters for each simulation problem, determined as a result of individual Bayesian sweeps. As determined using a multivariate importance score calculation over all hyperparameters, red denotes the most sensitive hyperparameter, and blue the second-most.

is surrounded by Robin-type boundary conditions, and the source is located outside of the domain. For each of the 12 problems demonstrated in Fig. 1, we perform Bayesian sweeps over the possible values specified in Table 1. This is because each PINN setup is highly sensitive to the unique Robin boundary conditions of each solution, requiring careful tuning of the training hyperparameters. A total of 50 training runs were performed in each of the 12 sweeps, over a sub-range of hyperparameters that was determined as most promising based on the results of a random-selection sweep of the SiO2 120 by 120 simulation problem consisting of 1200 training runs.

The results of the Bayesian hyperparameter sweeps are recorded in Fig. 2, which illustrate the challenges related to training domain-specific PINNs on general, heterogeneous simulation domains. Convergence is highly sensitive to optimization parameters, but not necessarily the same parameters, depending on the simulation problem. As demonstrated in Fig. 2,

there is no discernable pattern regarding the best-performing set of hyperparameters across simulation problems. However, as illustrated in Fig. 2, the most common hyperparameter of high importance to the outcome of the sweep (as determined by the multivariate "importance" calculation Liu & Motoda (1998)), is the boundary condition weight (i.e., the weighing factor between the bulk loss and the boundary condition loss during training). The sensitivity of the performance of the model to select hyperparameters, and the large variety of parameters from problem to problem, results in a laborious training process when applying PINNs to general simulation domains.

A general trend emerges in the simulation sweep, illustrated in Fig. 1: as the material refractive index increases (and with it the complexity of the field profile), and the simulation domain size scales up, the performance of the PINN rapidly deteriorates. Although the PINN performs reasonably well for smaller domain sizes with domains consisting of relatively lower refractive index materials, the fully connected model PINN was incapable of scaling to the larger domain sizes solved by SNAP-DDM with the same levels of accuracy.

The physics training aspect of domain-specific solvers is closely related to the training process in operator learning. There is significant progress in physics-informed operator solvers. Physics-informed neural operators (PINOs) for example, rely on the FNO framework to learn the function-mapping operator by training on both data and PDE constraints at different resolutions. Li et al. (2023) Although it is demonstrated to work well for heterogeneous simulation domains and has several interesting properties, such as training on lower-frequency problems and generalizing to higher-frequency sources, PINO is implemented using the FNO as a backbone, which results in difficulties in scaling to higher dimensions. Li et al. (2023) PIDeepONets Wang et al. (2021) is another example of momentous progress in physics-informed operator learning, which biases the output of DeepONets Lu et al. (2021) towards physically robust solutions. PIDeepONets augments DeepONets by using automatic differentiation over the input variables, similar to PINNs. Wang et al. (2021) Although these models demonstrate impressive solution accuracy improvements, generalizability, and data efficiency, the models are computationally expensive to train because the training dataset size is a product of the number of input functions and evaluation coordinates. Wang et al. (2021) The training computational complexity is further complicated by the significant computational graph size increase due to the automatic differentiation of the input parameters, resulting in significantly longer training time compared to DeepONets. PIDeepONets was not able to converge when trained on the Helmholtz equation for the H field, although this can likely be ameliorated by increasing the number of training collocation points and careful tuning of the training hyperparameters with sufficient compute availability.

## S2 FOURIER NEURAL OPERATOR ON THE FULL PROBLEM

In this section, we demonstrate the results and difficulties in training an SM-FNO model on the full-sized problem consisting of 960 by 960 pixels. The training data consists of a total of 1000 data samples, which is split into 900 training samples and 100 test samples. Each input device consist of a random grayscale dielectric material map, a map of 4 line sources with a random profile, and a PML map produced by a 40-pixel thick UPML on four sides, which is constant.

We trained an SM-FNO model with 6 layers, 64 channels and 16 Fourier modes. The model has 262M weights and 531G FLOP per input device. The Adam optimizer is used with learning rate starting at 3e-4 and annealed to 1e-5 over 100 epochs. We used an NVIDIA RTX A6000 GPU with 48GB memory, and could fit up to a batch size of 4 during training.

From the training curve and sample visualizations, it is clear that the model is able to overfit to the training data and learn the lower frequency spectrum of the fields, but fails to generalize to test data. This is expected as we have seen that we need more than 100k data even for a 64 by 64 pixels subdomain with heterogeneous material and arbitrary boundary, it would only require orders of magnitude more data to learn similar problems on a larger scale.

We believe theoretically it is possible that with sufficient resource and time, an end-to-end model could be trained for a large problem. As a quick comparison, it takes about 2 hours

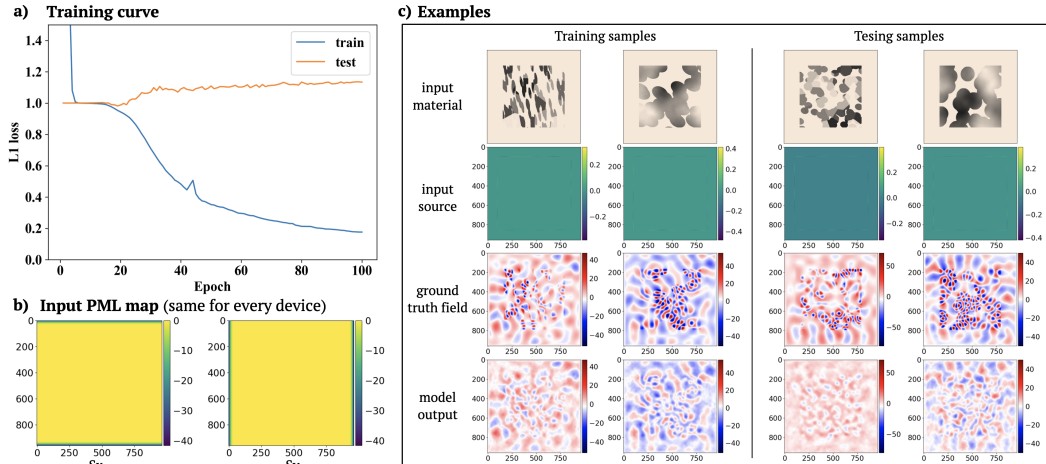

Figure 3: SM-FNO trained on full-sized problem with 960 by 960 pixels. a) Training curve for 100 epochs. b) The input PML map shared for each device. c) Samples from the training and testing datasets.

to generate the 1000 training samples on a desktop with 40 CPU cores. It would take over 2 months to generate 100k data. The scaling would be even worse for problems in 3D.

At the same time, we have demonstrated that cropping the same 1000 simulations to create subdomain dataset with size on the order of 100k to 1M could be sufficient for building semi-general subdomain solvers that is capable of solving a broad range of problems.

Other methods could be beneficial in practice like generating low-resolution data and use physics to help formulate high-resolution solutions (Li et al., 2021). We note that for wave-like problems, a minimum number of points needs to be sampled per wavelength to avoid aliasing, which sets the lower bound of problem complexity. Besides, end-to-end methods are usually designed for a fixed physical domain size, while the DDM approaches have the flexibility to be applied to different sized problems.