# OpenReview forum: "Towards general neural surrogate PDE solvers with specialized neural accelerators"
_ICLR.cc/2024/Conference — Submitted to ICLR 2024_

### Official Review · Reviewer_PJmS · 2023-10-30

**Soundness:** 2 fair
**Presentation:** 2 fair
**Contribution:** 1 poor
**Rating:** 5
**Confidence:** 3

**Summary:**

Neural networks can be used to learn the solution operators for PDEs. The more complex the PDE problem (in terms of initial data etc.) the harder it is to learn solution operators applicable to a wide class of problem instances. The authors propose to combine the ideas of domain decomposition methods with that of Fourier neural operators. They demonstrate how to successfully learn the solution operator for smaller sub-domains for problems that can be solved by iterating on a domain decomposition by updating the boundary condition based on neighboring solutions. The authors demonstrate that their learned neural operators can solve example problems in electromagnetics (magnetic field wave equation) and fluid mechanics (incompressible Navier-Stokes equation) via domain decomposed iteration. To successfully learn the solution operator the authors propose to add residual connections and a self-modulation mechanism to the Fourier neural operator architecture from the literature and demonstrate their usefulness in ablation experiments. Due to the iterated nature of the author’s proposed solution approach the solution is not one-shot and the computational cost increases in a non-trivial way as a function of problem size. Overall, it is not clear that using the proposed SNAP-DDM approach outperforms a conventional finite difference frequency domain solver.

**Strengths:**

- The authors present an original idea of applying the neural operator approach to domain decompositions which should be easier to learn systematically due to keeping them to a smaller sub-domain which should hopefully contain less complexity.
- The authors contribute architectural improvements to FNOs and demonstrate the usefulness to the example problems that they consider. These ideas seem likely to be useful to other applications of the FNO architecture as well.

**Weaknesses:**

- The evaluation setup is not very clear to me in terms of what was done on training data and what was done on evaluation data. For example, the 99% accuracy made in the introduction seems to be on training data (Table 1)? It makes sense to me that we want to understand how well in general the architecture can fit a neural operator if the training data for that specific problem instance is provided, but that does not tell me anything about how well the approach generalizes. My understanding is that Section 3.2 uses the previously trained neural operators on a held out problem instance (Figure 4). I think this distinction and implications about generalization need to be discussed more clearly. Especially, I'm wondering how dissimilar the instances can become and still converge or how does out-of-domain-ness impact convergence speed.
- Related work and background of how the contribution ties into the field is explained only on a surface level (in the introduction). Literature cited is mostly from numerical simulation publication venues and not machine learning. Lots of terminology that is used without definition will not be familiar to the broader ICLR audience, e.g. TE and TM polarization not necessarily familiar to non-physics audience, Bloch phase, finite difference frequence domain solvers. Furthermore, the insights gained in the paper are mostly useful to the numerical simulation community in a specific application area. Not sure how generally useful the contribution is to machine learning field because no new machine learning concept are introduced. So I am not sure the broader ICLR audience is the right one for this paper.

**Questions:**

- Since neural operators are approximations whose approximation error is hard to bound in advance, how is it possible how useful they will be for a given problem instance? Is there some way of combining them or using them to warm-start or speed up a traditional numerical solver with theoretical convergence guarantees (more like a pre-conditioner)?
- When can DDM methods be applied in principle / how general are they? Do they only lend themselves to the type of PDEs considered in the paper or are they much broader?
- Is the setup used in the paper (2D 960x960 grid) relevant to praciticioners? Can useful real-world problems be solved at that resolution?
- c1 and c2 be set to 1 for "simplicity" -> is this based on any sort of hyper param experiment? Technically, you also only need one of these hyper params, since the other one is implicit in alpha.
- For the time complexity measurements: Mean absolute accuracy of 15% seems like far from convergend (less than first 16 iterations in Figure 4), why was this chosen?
- Nit: I was a bit confused by the use of "pixels" terminology in a paper about PDE solutions

---

> ### Author Response · Authors · 2023-11-23
> **Thank you for your review!**
>
> >4.1. The evaluation setup is not very clear to me in terms of what was done on training data and what was done on evaluation data. For example, the 99% accuracy made in the introduction seems to be on training data (Table 1)? It makes sense to me that we want to understand how well in general the architecture can fit a neural operator if the training data for that specific problem instance is provided, but that does not tell me anything about how well the approach generalizes. My understanding is that Section 3.2 uses the previously trained neural operators on a held out problem instance (Figure 4). I think this distinction and implications about generalization need to be discussed more clearly. Especially, I'm wondering how dissimilar the instances can become and still converge or how does out-of-domain-ness impact convergence speed.
>
> All the table and plot data are conducted on 10k newly generated unseen test data. We have also added text in section 3.1 as well as in the titles of Tables 1 and 2. We have also added the dataset split and training details in section 3.1.
>
> We expect the performance of the model will deteriorate as the test data drift away from the training data distribution. For example for material dielectric constant outside of the range considered (epsilon = 1 to 16), or with different Reynolds numbers. The claim of our method emphasizes more on how to create random dataset (detailed in Appendix A) to enable model solving a broad range of problems within its distribution that’s sufficient for practical applications (shown in Figure 4), and we don’t claim that the method would extrapolate perfectly to unseen distributions.
>
> >4.2. Related work and background of how the contribution ties into the field is explained only on a surface level (in the introduction). Literature cited is mostly from numerical simulation publication venues and not machine learning. Lots of terminology that is used without definition will not be familiar to the broader ICLR audience, e.g. TE and TM polarization not necessarily familiar to non-physics audience, Bloch phase, finite difference frequence domain solvers. Furthermore, the insights gained in the paper are mostly useful to the numerical simulation community in a specific application area. Not sure how generally useful the contribution is to machine learning field because no new machine learning concept are introduced. So I am not sure the broader ICLR audience is the right one for this paper.
>
> We understand this work covers a broad range of topics and that there is some technical electromagnetic jargon in the manuscript, which we have attempted to limit with additional edits throughout.  The point of this manuscript is to simply use electromagnetics as a testbed for solving PDEs with neural networks, which itself is of interest to the machine learning community (the original FNO and PINNs papers already have thousands of citations from the applied mathematics and machine learning communities).  In this context, we push back on the idea that “no new machine learning concept is introduced.”  We introduce improvements to the FNO architecture (again, the initial paper is cited 1000+ times largely by machine learning specialists) that can enable improved solving of PDEs, show that FNO subdomain solvers can be utilized in a DDM framework, and we provide data generation schemes and physics-augmented training schemes that enable subdomain solvers operating with near unity accuracy.  All of these insights are not limited to electromagnetics or fluids but to any steady state PDE problem, making our insights in machine learning of broad interest to members in the computational, physical sciences, and data science communities.
>
>
> >4.3. Since neural operators are approximations whose approximation error is hard to bound in advance, how is it possible how useful they will be for a given problem instance? Is there some way of combining them or using them to warm-start or speed up a traditional numerical solver with theoretical convergence guarantees (more like a pre-conditioner)?
>
> Part of what we wanted to explore was whether we could solve PDE problems using only neural networks.  PDE solutions that > 90% accurate, which is what we have with SNAP-DDM, are sufficient for performing inverse design and optimization in fields such as nanophotonics. [1,2]  With that said, we agree that the idea of using neural operators as a means to find better initial guess or to be used as pre-conditioner to accelerate traditional solvers with theoretical convergence guarantees is a promising avenue for future research.
>
>
>
> [1] Zhelyeznyakov, Maksym, et al. "Large area optimization of meta-lens via data-free machine learning." Communications Engineering 2.1 (2023): 60.
>
> [2] Chen, Mingkun, et al. "WaveY-Net: physics-augmented deep-learning for high-speed electromagnetic simulation and optimization." High Contrast Metastructures XI. Vol. 12011. SPIE, 2022.

---

> > ### Author Response · Authors · 2023-11-23
> > **Continued response**
> >
> > >4.4. When can DDM methods be applied in principle / how general are they? Do they only lend themselves to the type of PDEs considered in the paper or are they much broader?
> >
> > DDM methods are a general iterative algorithm that are applicable to linear and nonlinear PDEs, elliptic, parabolic, and hyperbolic equations, as well as to problems with complex geometries and heterogeneous materials, with the theories most developed for elliptic PDEs. DDM for Maxwell’s equations is actually very challenging to solve due to the indefinite and ill-conditioned nature of the problem. Most of the demonstrations we had in the paper can directly apply to simpler problems including the Helmholtz equation, Possion equation, Laplace equation, etc.
> >
> > >4.5. Is the setup used in the paper (2D 960x960 grid) relevant to practitioners? Can useful real-world problems be solved at that resolution?
> >
> > We thank the reviewer for the consideration of practical usage of our method. First we want to point out that 960 by 960 grid is the size used in generating the 1000 training data. Our method could in principle be applied to arbitrary domain size (although for neural surrogate solvers, the DDM accuracy will decrease as problem goes too big). In the updated Figure 4, we have indicated the number of pixels in each demonstration, which we have shown up to a 2400 by 2400 grid size. The examples chosen include real-world, useful devices like microlens, metasurfaces, and thin-film stacks.
> >
> > From our experience,many  practical applications often utilize grid sizes ranging from 30x30 to 1000x1000 so the proposed method is well suited for many practical applications.
> >
> > >4.6. c1 and c2 be set to 1 for "simplicity" -> is this based on any sort of hyper param experiment? Technically, you also only need one of these hyper params, since the other one is implicit in alpha.
> >
> > We initially used c1 and c2 for ease of understanding. We agree that leaving one constant is enough for the relative strength of the two. We have updated the text with one constant c. The reason we left it as 1 is because we found the model performance is insensitive to its value between 0.1 and 10, which we also added in text. This is understandable because the model mainly relies on data loss for gradient descent, and uses both inner loss and boundary loss similar to a regularization term for more physical output.
> >
> > >4.7. For the time complexity measurements: Mean absolute accuracy of 15% seems like far from converged (less than first 16 iterations in Figure 4), why was this chosen?
> >
> > We appreciate the reviewer’s question on DDM accuracy. We have stated and cited in text that an accuracy of 15% is much better compared to for other state of the art neural surrogate solvers in photonics.  An additional reason why we choose 15% is because in order to run a benchmark for time complexity, we have experiments that run up to 2100 by 2100 pixels. We have discovered that 15% is about the threshold that the SNAP-DDM algorithm could consistently achieve for large problems in this size, which is how we choose it.

---

### Official Review · Reviewer_scWL · 2023-10-31

**Soundness:** 3 good
**Presentation:** 2 fair
**Contribution:** 3 good
**Rating:** 6
**Confidence:** 3

**Summary:**

This paper proposes SNAP-DDM, a method that utilizes deep learning in the context of domain decomposition methods. Each subdomain inside a given DDM framework is separately solved with a neural network-based solver and stitched together using appropriate boundary conditions. Depending on the contents of each subdomain, specialized neural operators are used. The main contribution of this work are modifications to the FNO architecture, in the form of residual connections inspired by the ResNet architecture and self-modulating connections inspired by transformers. The authors evaluate the proposed method on an electromagnetic and a fluid flow problem.

**Strengths:**

I see two main strengths of this paper, mainly in terms of originality and significance:

First, to me the application of established PDE deep learning methods to DDM appears to be an original and promising direction, even though I am not an expert in the domain that this paper targets. Nevertheless, I am still unsure if this paper would not be a better fit in a physics journal, as especially Section 2 heavy relies on physical details in the current version of this work.

Second, if the modifications to FNO hold in a more general setting, the proposed residual and self-modulating connections would be a valuable addition to the neural operator toolbox. Especially so, considering the success of ResNets and transformers compared to previous network architectures in the vision and language modeling domains.

Furthermore, source code is provided alongside this submission, which should help to improve the reproducibility of the shown results. However, I did not run or investigate the source code.

**Weaknesses:**

This paper appears to be unfinished in various aspects, and several presentation issues and lacking details make it difficult to clearly understand the methodology and judge the presented empirical results.

### Presentation

**P1:**
Even though there are a range of references in the introduction, in my opinion this paper would clearly benefit from a dedicated related work section and a more generic introduction. Otherwise connections relative to prior work are difficult to draw, especially for non-experts on the overlap area of DDMs, neural operators, and machine learning like myself.

**P2:**
There are statements throughout the paper that are vague, need further explanation, or require citations. Some examples are the following, but this is not an exhaustive list:
- *"but the are largely limited to systems featuring predetermined problem sites or fixed PDE parameters”* (abstract)
- the spectral bias issues of PINNs (end of second paragraph in section 1)
- the problems of current neural operator methods *“While much progress have been made… resource consuming and undesirable.”* (at the bottom of page 1 / the top of page 2)
- *“Furthermore, the training of separate networks… without loss of generality.”* (at the top of page 4)

**P3:**
The overall structure of the paper and especially the presentation of the methodology should be improved. It is not clear how exactly the network is used or what its outputs are, even at the end of the methodology section. Furthermore, the methodology section is not written in generic terms of the method, but simply as an example description on the electromagnetic experiment. For instance, it is not discussed how to choose networks in the generic case, or how SNAP-DDM works on the fluid flow problem.

**P4:**
The paper generally lacks polishing and contains a range of smaller issues and typos, for example:
- Number of abbreviations is relatively large and can easily become confusing
- Fig. 3/4/6: In my opinion, it is not ideal to show ground truth field and error in same colormap
- Fig. 4: Colorbars are missing
- Fig. 4/6/7: Subfigures are plotted inconsistently
- Fig. 7: What are the differently colored lines here? To me that was not clear from the description. Furthermore, the fontsize is too small
- Section 1, paragraph 2: “ansartz”
- Abstract: “near unity accuracy” (unclear and unusual formulation)
- Top of page 3: “Yee formalism (cite)”
- Begin of Section 4: “with L the being number of layers”


### Evaluations
**E1:**
The proposed changes in this work are first and foremost improvements to FNOs and not DDM. It is not clear why they are only tested on the DDM case, which makes things unnecessarily complicated. Instead, a direct comparison against FNOs and other baselines on full-sized PDE problems would be much more logical as a first step. With the current scope it is unclear if the observed improvements hold more generally.

**E2:**
It would be nice to have a comparison to a non-DDM model as a baseline, to get an idea of the achieved performance level. For example, a direct prediction via a fully convolutional neural network trained on different domain sizes comes to mind.

**E3:**
The physical problems are relatively limited, especially the fluid flow problem (where it is even unclear how SNAP-DDM works, as mentioned in P3 above). An interesting case would be a more complex fluid problem, like an unsteady flow. This would requires multiple time iterations (in addition to the solver iterations), but show that DDM is robust to temporal rollouts, which is a highly desirable property.

**E4:**
It is not mentioned how the data sets are split. This is especially relevant for Tab. 1 and 2, as it is unclear what is actually shown here, training loss, validation, or test performance? Furthermore, it would be interesting to see how SNAP-DDM performs for more complex evaluation tasks (slightly) outside of the training domain, for example different Reynolds numbers in the flow experiment. Or boundaries created by a fundamentally different generation method.

**E5:**
Tab 1 shows that more training data substantially improves model performance. Why are the other baselines not evaluated with the same amount of data for a fair comparison at the computational limit that would be used in practice as well?

**E6:**
Training details of the baselines are missing, and the appendix only contains a very rough overview of some parameters of each baseline. Furthermore, how are the most important hyperparameters for the baselines chosen?

**E7:**
While the proposed changes are promising compared to FNOs, it seems that other simpler architectures like Unet can achieve similar performance. Especially with recent Unet modernizations commonly used for diffusion models, even better results might be possible (see *“Denoising diffusion probabilistic models”* by Ho et al., NeurIPS 2020, or *“Diffusion models beat GANs on image synthesis”* by Dhariwal and Nichol, NeurIPS 2021).

### Summary
Overall, this work feels unfinished and in my opinion needs a larger revision in terms of presentation and more rigorous evaluations. Due to the chosen problem setup inside a DDM solver, it is also difficult to tell if the improvements to the FNO architecture actually hold in a more general setting. This leads to my overall recommendation of reject for the current state of this paper.

### Update after Author Response
With the improved presentation and additional results, the insights from combining DDM with FNOs are certainly an interesting direction. Nevertheless, I think this paper would clearly benefit from further improving the presentation and investigating the strengths of the other baseline variants, especially U-Net in more detail. Finally, showing the benefits of the proposed improvements to FNOs in a more general setting would be a useful addition. As a result, I reconsidered my original evaluation of this work and updated my original review with the following changes:
- Soundness Score: increased from *2 fair* to *3 good*
- Presentation Score: increased from *1 poor* to *2 fair*
- Overall Score: increased from *3 reject* to *6 marginally above the acceptance threshold*

**Questions:**

**Q1:**
How can the data error computed via an L1 loss in Fig. 3/6 be negative? According to equation 4 it is just a sum over the absolute difference which should always be positive?

**Q2:**
What are the results when evaluating the best baseline (i.e. Unet) in the same way as shown for SNAP-DDM in Fig. 4/6b?

**Q3:**
I did not fully understand the choice for the ablation study models in Tab. 3, especially w.r.t. changing the architecture at the same time as the model sizes $L$ and $C$. The current presentation makes it unclear if the differences are due to architectural changes or the model size.

**Q4:**
On a rather abstract level, the iterative refinement approach of the solver has some interesting connections to diffusion models, that also iteratively refine an initial prediction. What are your thoughts on the usage of diffusion models within DDMs: Do you see some potential to map the solver iterations to an iterative model training schedule, achieving a physical diffusion-style DDM framework?

---

> ### Author Response · Authors · 2023-11-23
> **Thank you for your review!**
>
> >3.1. Even though there are a range of references in the introduction, in my opinion this paper would clearly benefit from a dedicated related work section and a more generic introduction. Otherwise connections relative to prior work are difficult to draw, especially for non-experts on the overlap area of DDMs, neural operators, and machine learning like myself.
>
> We have rewritten parts of the introduction in an attempt to add clarity to the current state-of-the-art in PDE solving and to further highlight our contributions.  With the page limit and the addition of other content to the main text, we were not able to add too much more text but hope that the additional cited references can further help an interest non-specialist learn more about these non-trivial subjects.
>
> >3.2. There are statements throughout the paper that are vague, need further explanation, or require citations. Some examples are the following, but this is not an exhaustive list:
> 1. but the are largely limited to systems featuring predetermined problem sites or fixed PDE parameters” (abstract)
> 2. the spectral bias issues of PINNs (end of second paragraph in section 1)
> 3. the problems of current neural operator methods “While much progress have been made… resource consuming and undesirable.” (at the bottom of page 1 / the top of page 2)
> 4. “Furthermore, the training of separate networks… without loss of generality.” (at the top of page 4)
>
> We have edited the entire document to modify statements that are vague or that require further explanation, and we have specifically addressed the examples cited.
>
> >3.3. The overall structure of the paper and especially the presentation of the methodology should be improved. It is not clear how exactly the network is used or what its outputs are, even at the end of the methodology section. Furthermore, the methodology section is not written in generic terms of the method, but simply as an example description on the electromagnetic experiment. For instance, it is not discussed how to choose networks in the generic case, or how SNAP-DDM works on the fluid flow problem.
>
> We have updated Figure 1 to indicate more clearly the inputs and outputs of each subdomain model and how these inputs and outputs work synergistically in the DDM framework.  We also have a dedicated section on boundary value update that visualizes how boundaries within a subdomain are extracted and used to update the boundary value of the nearest neighbors.  While this example and flow chart is indeed focused on an electromagnetics problem, we believe it is now posed in a sufficiently generic manner as to be readily adapted to other types of physics problems.  We have also updated the caption of Figure 1 to indicate that the same principle (boundary models, source models and material models) could be applied to general PDE problems, including fluid problems.
>
> >3.4. The paper generally lacks polishing and contains a range of smaller issues and typos, for example: …
>
> We have fixed the typos and added full clarified abbreviations.

---

> > ### Author Response · Authors · 2023-11-23
> > **Continued response**
> >
> > >3.5. The proposed changes in this work are first and foremost improvements to FNOs and not DDM. It is not clear why they are only tested on the DDM case, which makes things unnecessarily complicated. Instead, a direct comparison against FNOs and other baselines on full-sized PDE problems would be much more logical as a first step. With the current scope it is unclear if the observed improvements hold more generally.
> >
> > >Related: 3.6. It would be nice to have a comparison to a non-DDM model as a baseline, to get an idea of the achieved performance level. For example, a direct prediction via a fully convolutional neural network trained on different domain sizes comes to mind.
> >
> > While we agree that the main neural network improvements in this study is on the FNO architecture, we maintain that the combination of surrogate subdomain solvers with DDM is itself a point of novelty because: 1) this hybrid algorithm has not been reported in the literature; 2) we introduce many innovations in getting this hybrid algorithm to work, including our specific data generation scheme (details in Appendix A), the choice with specific boundary condition (transparent Robin type), and the use of  data+physics subdomain model training; and 3) our enhanced FNO architecture for the subdomain solvers is required to get DDM to accurately converge.  Ultimately,  the proposed SNAP-DDM scheme is able to do inference on arbitrary sized simulation domains, as demonstrated in Figure 4, which is a new feature in neural network-enhanced PDE solving.  We believe that our algorithmic framework combining surrogate subdomains with DDM provides a general and new pathway for solving PDEs and that it will lead to new innovations at the intersection of machine learning and DDM.
> >
> > With that in mind, we agree that benchmarking our results with the solving of full-sized problems using other end-to-end machine learning algorithms is prudent and we add the following to the SI:
> > - We have conducted extensive architecture, activation function, and parameter search from multiple PINN models and reported our findings. We find that it remains challenging to scale current strategies to large domain sizes and heterogeneous materials with high dielectric constant.  These findings are consistent with other works in the literature.
> > - We also report the findings for training a neural operator on the entire domain (960 by 960 pixels) in the SI. We observe that the model size, data required for generalization, and the number of Fourier modes required quickly becomes intractable with larger domains. We also want to note that it still remains non-trivial to have a scheme work for arbitrary domain sizes, which is one key benefit of our DDM-based approach.
> >
> > In summary, our extensive experiments on full domain problems revealed that networks like FNOs struggle with large-scale PDEs due to prohibitive memory and computational requirements. This finding underscores the relevance of DDM as a more feasible and efficient framework for handling such complex problems. Our approach aims to push the boundaries of what's possible in PDE solving, and the observed improvements in the DDM context are indicative of the potential and scalability of our proposed changes. We believe this focus offers a more realistic and applicable contribution to the field, addressing the limitations of current methods when dealing with large-scale, real-world problems.
> >
> > >3.7. The physical problems are relatively limited, especially the fluid flow problem (where it is even unclear how SNAP-DDM works, as mentioned in P3 above). An interesting case would be a more complex fluid problem, like an unsteady flow. This would require multiple time iterations (in addition to the solver iterations), but show that DDM is robust to temporal rollouts, which is a highly desirable property.
> >
> > In response to the concerns raised about the limited scope of physical problems addressed, particularly fluid flow, we would like to clarify the primary objective of our work. The focus of this research is not to extensively explore specific fluid dynamics problems, but rather to demonstrate the advancements in our proposed framework that pairs a more advanced Fourier Neural Operator (FNO) with Domain Decomposition Methods (DDM) for steady state problems. This combination is specifically designed to tackle large domain Partial Differential Equation (PDE) problems effectively. For many fields like those in electromagnetics, heat transfer, and acoustics, steady state PDE solvers are broadly interesting and useful.
> >
> > While exploring complex fluid problems like unsteady flow is an intriguing future research direction, it falls outside the immediate scope of this study. The complexity in generating datasets for such narrowly defined and specialized fields would not only divert resources but also shift the focus away from the core contribution of our work.

---

> > > ### Author Response · Authors · 2023-11-23
> > > **Continued response**
> > >
> > > >3.8. It is not mentioned how the data sets are split. This is especially relevant for Tab. 1 and 2, as it is unclear what is actually shown here, training loss, validation, or test performance? Furthermore, it would be interesting to see how SNAP-DDM performs for more complex evaluation tasks (slightly) outside of the training domain, for example different Reynolds numbers in the flow experiment. Or boundaries created by a fundamentally different generation method.
> > >
> > > All the table and plot data are conducted on 10k newly generated unseen test data, which we have added text in section 3.1 as well as in the titles of Tables 1 and 2. We have also added the dataset split and training details in section 3.1.
> > >
> > > We expect the performance of the model will deteriorate as the test data drift away from the training data distribution, for example, for material dielectric constants outside of the range considered (epsilon = 1 to 16) or with different Reynolds numbers. As such, our method emphasizes on how to create random datasets (detailed in Appendix A) to enable model solving over a broad range of problems within its distribution.  This is  sufficient for practical applications, shown in Figure 4.  It would be a very interesting future research direction to explore and understand the extrapolation capabilities of these models.
> > >
> > > >3.9. Tab 1 shows that more training data substantially improves model performance. Why are the other baselines not evaluated with the same amount of data for a fair comparison at the computational limit that would be used in practice as well?
> > >
> > > We agree that training on 1M data makes for a fairer comparison. We have added new benchmark in Table 1 and Figure 3 with models trained on 1M data (except for the Swin Transformer, which takes 2 months to complete and is not ideal in practice).
> > >
> > > >3.10. Training details of the baselines are missing, and the appendix only contains a very rough overview of some parameters of each baseline. Furthermore, how are the most important hyperparameters for the baselines chosen?
> > >
> > > We thank the reviewers for this request. We have added training details in section 3.1.
> > > The details for architectural parameters and training for the Swin Transformer model are included in the updated Appendix F, where we selected the parameters based on the same models used for semantic segmentation and adjusted to suit our subdomain problems.
> > >
> > > >3.11. While the proposed changes are promising compared to FNOs, it seems that other simpler architectures like Unet can achieve similar performance. Especially with recent Unet modernizations commonly used for diffusion models, even better results might be possible (see “Denoising diffusion probabilistic models” by Ho et al., NeurIPS 2020, or “Diffusion models beat GANs on image synthesis” by Dhariwal and Nichol, NeurIPS 2021).
> > >
> > > We thank the reviewer for mentioning improved UNet architecture used in diffusion models. As we added the 1M dataset benchmark, we observe that the proposed SM-FNO architecture achieves much lower loss then the UNet and other architectures, even with fewer number of weights and half the FLOP count.
> > >
> > > While we have done parameter and architecture fine tuning to optimize the current baseline models, we agree that more engineering effort could be applied to each of the baseline models for better performance. We are not claiming that improved FNO architecture is the only option to be applied to a DDM setup, but rather that these architectural and loss function improvements enables existing models to achieve the level of accuracy needed in a DDM setup, which of course, applies to other architectures as well.
> > >
> > > >3.12. How can the data error computed via an L1 loss in Fig. 3/6 be negative? According to equation 4 it is just a sum over the absolute difference which should always be positive?
> > >
> > > We thank the reviewer for pointing out this mistake. We have changed the label to be “Relative field error”, which is the real-valued field difference that we aimed to show.
> > >
> > > >3.13. What are the results when evaluating the best baseline (i.e. Unet) in the same way as shown for SNAP-DDM in Fig. 4/6b?
> > >
> > > We have indicated in Figure 8 that using a less accurate subdomain model would greatly affect the SNAP-DDM algorithm accuracy, and we use it to indicate that UNet and other FNOs trained on 1M data are not sufficiently accurate to be used in a DDM setup. Thus we decide not to add experiment to compare DDM setups using these models trained on 1M data.
> > > We are not eliminating the possibility that with more engineering effort, and as a future research direction, it would be interesting to explore how well an optimized UNet could perform in practical DDM applications.

---

> > > > ### Author Response · Authors · 2023-11-23
> > > > **Continued response**
> > > >
> > > > >3.14. I did not fully understand the choice for the ablation study models in Tab. 3, especially w.r.t. changing the architecture at the same time as the model sizes and the current presentation makes it unclear if the differences are due to architectural changes or the model size.
> > > >
> > > > For each of the architectures, we did a parameter sweep on the number of layers, channels and Fourier modes, within a constrained number of parameters and FLOP counts to enable a fair comparison, to select best performing ones. The point of the ablation study is to ultimately show that both modifications are necessary for training an accurate and expressive deep SM-FNO model.
> > > >
> > > > >3.15. On a rather abstract level, the iterative refinement approach of the solver has some interesting connections to diffusion models, that also iteratively refine an initial prediction. What are your thoughts on the usage of diffusion models within DDMs: Do you see some potential to map the solver iterations to an iterative model training schedule, achieving a physical diffusion-style DDM framework?
> > > >
> > > > We thank the reviewers for bringing up this very interesting connection. Below are our ideas:
> > > >
> > > > –While both are iterative processes, we see fundamental differences that exist. The error updating mechanism in DDMs cannot be simplistically equated to a noise model as used in diffusion processes. The iterative refinements are inherently distinct from the stochastic and probabilistic nature of noise models in diffusion processes. Especially for electromagnetic problems, the fundamental wave propagation process is very different from the noise diffusion process. It might have a closer relation to fluid motion or heat transfer systems.
> > > >
> > > > –If the diffusion model is applied on the full-sized problem, the challenge for scaling still remains as we have demonstrated in the SI. It might have a connection to a learnable boundary update algorithm for faster convergence, together with other subdomain solver/models.
> > > >
> > > > –If the diffusion model is applied at the subdomain level, we don’t see the need for an iterative solver for a subdomain problem as of yet.
> > > >
> > > > –We also want to note that other iterative algorithms might have connection to diffusion model, and be of interest to the reviewer, for example conjugate gradient methods or krylov subspace methods.

---

### Official Review · Reviewer_mta4 · 2023-11-01

**Soundness:** 3 good
**Presentation:** 1 poor
**Contribution:** 3 good
**Rating:** 5
**Confidence:** 3

**Summary:**

The paper introduces a method that employs domain decomposition techniques to help solve steady-state partial differential equations PDEs, for which known physical prior information is available. The model operates by mapping the parameter functions of the equation to its steady-state solution. This is achieved through the division of the entire domain into multiple subdomains. Notably, the surrogate models are exclusively trained using data from the subdomains' results and boundary conditions. The PDE is then solved iteratively, incorporating a physical loss, and updating the boundary conditions for each subdomain. The results of the proposed model are promising.

**Strengths:**

- The paper investigates the utilization of an existing physical model to streamline the solving of partial differential equations (PDEs) by spatially decomposing the domain. This approach reduces the complexity of the required model for each subdomain. This offers insights into incorporating additional physical priors to enhance the scalability of current neural PDE surrogate models.
- The method is evaluated on various electromagnetic systems with diverse domain shapes. The modular design of the model makes it applicable for solving the same PDE in different settings.

**Weaknesses:**

- In terms of presentation quality, I recommend enhancing the overall structure of the paper by providing a more visible and distinct description of the model, including a concise definition of the input and the output. Additionally, it would be beneficial to separate and clarify the sections for the model, training, and evaluation processes. Currently, understanding the entire training pipeline is challenging as it is intertwined with the model description, the existing algorithm, and the experimental results. Furthermore, I suggest reorganizing the presentation of experimental results with a clearer structure, dedicating specific sections to data preparation, technical details, and result analysis.
- Baselines comparison: While the model is currently benchmarked against baseline architectures for subdomain training, it is essential to consider a broader comparison with physics-informed baselines such as PINNs and PINOs/PiDeepONet on the entire domain. This extended evaluation would provide a more comprehensive understanding of how the proposed method compares to established DL-based physics-informed techniques.
- Regarding terminology: I was somewhat surprised when I noticed that the authors introduced an additional residual connection alongside the one that already existed. (Please note that point-wise convolution is equivalent to applying a linear layer to each pixel and is a conventional implementation of residual connections when the input and output channel counts do not match.) Have the authors considered replacing the point-wise convolution with a simple identity addition?

**Questions:**

- It's quite unexpected that the original FNO struggles to predict the outputs of not-so-complex subdomains accurately. It would be valuable if the authors could provide additional insights into the specific challenges or limitations the original FNO faces when applied to these cases.
- In Figure 7, it's intriguing to observe the solver's convergence with iterations, especially considering that the subdomain models are trained solely with the ground truth subdomain boundary conditions. To gain a deeper understanding of this convergence behavior, it would be beneficial to include a more detailed graph that provides further insights into the solver's performance throughout the entire testing process. The current figures show data from only five selected points, but a more precise graph could shed light on the solver's convergence across the entire testing period.

---

> ### Author Response · Authors · 2023-11-23
> **Thank you for your review.**
>
> >2.1 In terms of presentation quality, I recommend enhancing the overall structure of the paper by providing a more visible and distinct description of the model, including a concise definition of the input and the output. Additionally, it would be beneficial to separate and clarify the sections for the model, training, and evaluation processes. Currently, understanding the entire training pipeline is challenging as it is intertwined with the model description, the existing algorithm, and the experimental results. Furthermore, I suggest reorganizing the presentation of experimental results with a clearer structure, dedicating specific sections to data preparation, technical details, and result analysis.
>
> We thank the reviewer for the critical suggestions on presentation.
> - We have updated Figure 1 with much clearer indication of inputs and output of each subdomain model, and we have a dedicated section on boundary value update that visualizes how boundaries are extracted, and update to overlapping nearest neighbors.
> - Due to the page limit, we have left details of data generation in Appendix A.
> - We added training details in section 3.1.
>
> >2.2 Baselines comparison: While the model is currently benchmarked against baseline architectures for subdomain training, it is essential to consider a broader comparison with physics-informed baselines such as PINNs and PINOs/PiDeepONet on the entire domain. This extended evaluation would provide a more comprehensive understanding of how the proposed method compares to established DL-based physics-informed techniques.
>
> We agree it is necessary to show the comparison on the entire domain. We have added two experiments in the SI:
> - We have conducted extensive architecture, activation function, and parameter search for multiple PINN models applied to simulation problems of varying domain sizes with heterogeneous material simulation domains, with material refractive index spanning from air to silicon. We provide extensive discussion in Section D (Domain-Specific Physics-Informed Neural Networks) in the SI.  We find that it remains challenging to scale current strategies to large domain sizes and heterogeneous material with high dielectric index.  This observation is consistent with other works in the literature.
> - We also report in the SI the findings for training a neural operator on a large scale domain (960 by 960 pixels). We observe that the model size, data required for generalization, and the number of Fourier modes required quickly becomes intractable with larger domains. We also note that even if enough resources and time make such large scale training possible, it still remains non-trivial to have a scheme work for arbitrary domain sizes, which is one key benefit of our DDM-based approach.
>
> >2.3. Regarding terminology: I was somewhat surprised when I noticed that the authors introduced an additional residual connection alongside the one that already existed. (Please note that point-wise convolution is equivalent to applying a linear layer to each pixel and is a conventional implementation of residual connections when the input and output channel counts do not match.) Have the authors considered replacing the point-wise convolution with a simple identity addition?
>
> As discussed in point 1.4 above, we have indeed verified that initializing the W matrix with identity matrix (plus a kaiming initialization), is equivalent to the default kaiming initialization plus a residual connection. We have added text in the manuscript to clarify this point.
>
> >2.4. It's quite unexpected that the original FNO struggles to predict the outputs of not-so-complex subdomains accurately. It would be valuable if the authors could provide additional insights into the specific challenges or limitations the original FNO faces when applied to these cases.
>
> We certainly agree that the subdomain size (64 by 64 pixels) and square shape are simple, but we also emphasize that the modeling of the highly heterogeneous and high contrast media (epsilon ranging from 1 to 16), together with our consideration of arbitrary boundary conditions, makes our problem very challenging to learn for any FNO.  To add further clarification, we have made the following modifications (all in Figure 3 and Table 1):
> - We applied the correct initialization for the original FNO model, which now performs better and makes for a fairer comparison.
> - We added a subdomain benchmark with 1M total data size and show that due to the heterogeneous grayscale material distribution and arbitrary robin boundary conditions, our dataset is indeed complex enough that large training data size is required for generalization.
> - We also include a benchmark against a recently improved FNO named “Factorized FNO” (which we cited in the first version) to strengthen our argument that input-dependent modulation is crucial in learning complex heterogeneous dataset.

---

> > ### Author Response · Authors · 2023-11-23
> > **continued response**
> >
> > >2.5. In Figure 7, it's intriguing to observe the solver's convergence with iterations, especially considering that the subdomain models are trained solely with the ground truth subdomain boundary conditions. To gain a deeper understanding of this convergence behavior, it would be beneficial to include a more detailed graph that provides further insights into the solver's performance throughout the entire testing process. The current figures show data from only five selected points, but a more precise graph could shed light on the solver's convergence across the entire testing period.
> >
> > We have updated Figure 7 with a more precise graph with smaller step sizes to show the convergence properties for different devices.

---

### Official Review · Reviewer_RuAk · 2023-11-09

**Soundness:** 3 good
**Presentation:** 3 good
**Contribution:** 3 good
**Rating:** 6
**Confidence:** 4

**Summary:**

# Initial comment
In this paper, the authors try to extend the current learning-based PDE surrogate solvers to handle problems with larger scale, more complicated boundaries, and varying parameters, by integrating domain decomposition methods. In each subdomain, the problem is solved by neural operators with extra residual connection and modulation encoders. Experiments on electromagnetic and fluidic flow problems are performed to demonstrate the effectiveness of the proposed model, compared with FNO, UNet, and Swin Transformer.

# After author-reviewer rebuttal
Overall, thanks very much for your detailed response, enhanced experiments, and modification of the manuscript.

- Thanks for confirming some of the points I provided in the previous review.
- Thanks for the clarification on how the boundary information is handled in the proposed method.

With all the factors considered, as well as the fact that a positive score is already given, I decide to remain my original score for now.

**Strengths:**

- This is one of the earlist works to combine Neural Operators and DDM.
- The motivation is clear and reasonable, as described at the end of the third paragraph in the Introduction Section.

**Weaknesses:**

- The proposed method will perform self-consistent iterations, introducing extra complexity and convergence issues.
- The scale and shape of the PDE problems considered in this paper are not complicated enough to be fully convincing to me.
- The design is relatively straightforward and inflexible, resulting in limitations such as fixed subdomain size and shape.

**Questions:**

- One contribution the paper claimed is that extra residual connection is added to the FNO module. But the original FNO block already contains a parameterized residual connection. From the description in the paper, I understand that it is validated by experiments. But what is the benefit of such design in principle? Specifically speaking, if the W matrix in FNO is initialized as an identity matrix, will there be any difference?
- I am still not very clear about how the boundary information is fed into SNAP-DDM. A bit more explanation will be appreciated.

---

> ### Author Response · Authors · 2023-11-23
> **Thank you for your comment**
>
> Thank you for your review.
>
> >1.1. The proposed method will perform self-consistent iterations, introducing extra complexity and convergence issues.
>
> The reviewer’s comment on convergence on self-consistent iterations is well-taken. DDM methods are mature iterative methods for solving large scale PDEs, especially elliptical PDEs, and are the basis for parallelizing the solving of PDEs on CPU servers. While there remain theoretical research questions pertaining to the convergence stability of DDM with Maxwell’s Equations, especially in the highly heterogeneous dielectric case, we do ultimately demonstrate stable performance with our method, as demonstrated in Figures 4 and 7 with large scale high dielectric contrast problems.  This type of performance is unprecedented for neural network-based PDE solving.
>
> >1.2.  The scale and shape of the PDE problems considered in this paper are not complicated enough to be fully convincing to me.
>
> We agree that the size (64 by 64 pixels) and square shape of our targeted subdomains are simple, but our targeted subdomain PDE problem is nonetheless complex and non-trivial because it is highly heterogeneous, contains dielectric constant media (epsilon = 16), and contains fully arbitrary boundary conditions.  Very few demonstrations in the literature involving domains of any size capture this level of complexity. We emphasize that while DDM is well known in the mathematics community, the utilization of a surrogate subdomain solver together with DDM has not been possible until this work due to the accuracy limitations in prior FNO algorithms.  We also make the following modifications to further elucidate the quality of our result (all in Figure 3 and Table 1):
> - We applied the correct initialization for the original FNO model, which now performs better and makes a fairer comparison.
> - We add a subdomain benchmark with 1M total data size, to show that due to the heterogeneous grayscale material distribution and arbitrary Robin boundary conditions, our dataset is indeed complex enough that large training data size is required for generalization.
> - We also include a benchmark against a recently improved FNO named “Factorized FNO” (which we cited in the first version) to strengthen our argument that input-dependent modulation is crucial in learning a complex heterogeneous dataset.
>
> >1.3. The design is relatively straightforward and inflexible, resulting in limitations such as fixed subdomain size and shape.
>
> We fully note that the subdomain problem size is fixed.  However, the whole point of SNAP-DDM is that we can ultimately tile these subdomain solvers in arbitrary ways to solve large scale problems of arbitrary size, as demonstrated in Figure 4.  This flexibility is a feature of SNAP-DDM and it exceeds the flexibility of all previous neural network surrogate solvers.
>
> >1.4. One contribution the paper claimed is that extra residual connection is added to the FNO module. But the original FNO block already contains a parameterized residual connection. From the description in the paper, I understand that it is validated by experiments. But what is the benefit of such design in principle? Specifically speaking, if the W matrix in FNO is initialized as an identity matrix, will there be any difference?
>
> We have verified that mathematically, and from our experiments, initializing the W matrix with identity matrix (plus a kaiming initialization), is equivalent to the default kaiming initialization plus a residual connection. We still decide to keep the residual connection in Figure 2 for clarity and we add the following text in Section 2: “We note that the residual connection is equivalent to initializing the 1x1 convolutional layer W using identity plus Kaiming or Xavier initialization, but we keep the residual connection in Figure 2 for clarity and ease of implementation.” We also updated Figure 3 and Table 1 to have this better initialization for the original FNO benchmark, which makes for a much fairer comparison. We leave the less ideal initialization in the ablation study to show that explicit residual connection (or initialization) is necessary to reliably train deep networks.
>
> >1.5. I am still not very clear about how the boundary information is fed into SNAP-DDM. A bit more explanation will be appreciated.
>
> The Robin boundary values are inputted into the subdomain models as images with 64x64 pixel dimensions that match the subdomain size.  Boundary values are located at the perimeter of the image and the rest of the pixels interior to the image are set to zero.  We have updated Figure 1 with much clearer indication of inputs and output of each subdomain model, and we have a dedicated section on boundary value update that visualizes how boundaries are extracted, and update to overlapping nearest neighbors.
>
> [1] Li, Zongyi, et al. "Fourier neural operator with learned deformations for pdes on general geometries." arXiv preprint arXiv:2207.05209 (2022).

---

### Author Response · Authors · 2023-11-23
**Summary Response**

We thank all reviewers for their insightful questions and constructive feedback. Key changes in the revised submission include:

- Thanks to the comments, we realized that the DDM framework, together with subdomain model inputs and outputs are not clearly illustrated in Figure 1.  We have redrawn the figure and it now features:

    - The inputs and output for each subdomain model are clearly shown

    - The boundary value update process is explained in detail.

- Several reviewers pointed out that the W matrix (1 by 1 convolution) in the original FNO architecture could serve as a residual connection.
    - We have tested and verified that equivalent to the residual connection, an explicit initialization of the W matrix is required to obtain stable deep training. We decide to still keep the residual connection in Figure 2 for clarity and ease of implementation but have added clarifying text delineating this equivalence.
    - Furthermore, we have updated Table 1 and Figure 3 with the original FNO architecture now adopting the correct initialization (equivalently, adding the residual connection) that results in better performance. We thank the reviewers and agree that this makes a fairer comparison.

- Figure 4 now uses centered colormap, and indicate each simulation size in pixels.

- Figure 6 is re-drawn with finer resolution.

- In response to the concern about complexity on the subdomain problems, we made these changes:

    - We have clarified that all data and plots in Tables and Figures are evaluated on unseen test data.

    - We added subdomain benchmark with a recent improved variant of FNO architecture, named Factorized-FNO, which we also cited in the original submission, to strengthen our argument that input-dependent modulation is crucial in learning complex heterogeneous dataset.

    - We added benchmark on models trained on 1M data to show that our dataset is indeed complex enough due to the heterogeneous grayscale material distribution and arbitrary robin bounadry conditions, thus large training data size is required for generalization. Through this we also show the proposed SM-FNO architecture scales well to large model and data sizes.
    - Ablation study is updated with these changes as well, together with updated losses with carefully finetuned parameters.

- In response to the concern about complexity on the full-sized problems, especially compared to end-to-end models like PINN and neural operators without DDM (even though they are trained for fixed domain size), we added experiments in the SI section:

    - To compare with PINN, we performed thorough model architecture, activation function and parameter sweep, and show that PINNs struggle to solve heterogeneous or high-frequency (high dielectric contrast) problems, as has been reported in other works. We discovered using sinusoidal activation function gives better performance, but still scales poorly to larger problem sizes and higher indexed materials.

    - We also attempted to train a full-size SM-FNO model that takes the full sized (960 by 960 pixels) dielectric, source and PML as input and learns the problem. Multiple challenges exist in the scaling of model complexity, number of Fourier model, and the amount of training data, which exceeded the limit of 48GB memory of the GPU we used. We obtained acceptable training loss (17%) but observed severe overfitting (~100% loss on test data), which indicates more training data is required which becomes impractical to generate.
We believe the experiments added demonstrated the advantage of our approach, which significantly reduces the difficulty in scaling up neural surrogate models.

---

### Meta-Review · Area_Chair_Rd1k · 2023-12-06

**Metareview:**

The paper proposes SNAP-DDM, a domain decomposition method (DDM) for solving PDEs using neural operators. The main contribution of this approach, in light of the large existing literature on neural PDE solvers, is to show that very large PDE problems can be solved by subdividing global boundary value problems into local problems, solving the smaller problems using specialized neural solvers (in this paper, the authors focus on FNO), and stitching together the solutions using DDM ideas. The method is validated on PDEs arising in two application domains: 2D electromagnetics and fluidic flow.

The reviewers generally appreciated the direction taken in this paper. The idea (of combining neural PDEs with DDM) is quite natural, and to the best of my knowledge, novel. The evaluations for the electromagnetics use case showed that the method is extendable to challenging domain shapes. Overall, the technical parts are not terribly surprising but constitute a solid contribution.

However, some reviewers found the presentation hard to follow. Multiple reviewers pointed out that the paper is a bit too heavy on the details on the underlying physics of the two specific use cases, while being too light on the ML details (apart from a couple of paragraphs surrounding Figure 2, there is almost no treatment -- mathematical or otherwise -- about the SNAP-DDM method itself, how the authors arrived at it, various design choices they considered).

This was a tough decision, but in the end I side with the latter assessment: I also found the paper hard to read (despite being very familiar with the scientific ML literature). In its current form, the paper might be perhaps better appreciated as a submission to a computational physics journal (such as JCP) as opposed to being presented to an ICLR audience. If the authors wish to resubmit a future competitive ML venue, I'd recommend considerably expanding Section 2.1, adding a lot more details about the modification to FNO (as well as perhaps similar modifications + results of other neural PDE approaches: CNO, wavelet NO, DeepONets, ...), considerably expanding their evaluation suites and comparing with other methods that deal with complex domains, etc.

**Justification For Why Not Higher Score:**

Reviewers pointed out some positive aspect but in the end, the presentation and suitability of fit to ICLR remains questionable.

**Justification For Why Not Lower Score:**

N/A

---

### Decision · Program_Chairs · 2024-01-16

Reject